# Identification of novel toxins associated with the extracellular contractile injection system using machine learning

Aleks Danov[1,6], Inbal Pollin[1,6], Eric Moon [ID][2], Mengfei Ho [ID][2], Brenda A Wilson [ID][2], Philippos A Papathanos[3], Tommy Kaplan [ID][4,5] & Asaf Levy [ID][1 ✉]

## Abstract

**Secretion systems play a crucial role in microbe-microbe or host-microbe interactions. Among these systems, the extracellular contractile injection system (eCIS) is a unique bacterial and archaeal extracellular secretion system that injects protein toxins into target organisms. However, the specific proteins that eCISs inject into target cells and their functions remain largely unknown. Here, we developed a machine learning classifier to identify eCIS-associated toxins (EATs). The classifier combines genetic and biochemical features to identify EATs. We also developed a score for the eCIS N-terminal signal peptide to predict EAT loading. Using the classifier we classified 2,194 genes from 950 genomes as putative EATs. We validated four new EATs, EAT14-17, showing toxicity in bacterial and eukaryotic cells, and identified residues of their respective active sites that are critical for toxicity. Finally, we show that EAT14 inhibits mitogenic signaling in human cells. Our study provides insights into the diversity and functions of EATs and demonstrates machine learning capability of identifying novel toxins. The toxins can be employed in various applications dependently or independently of eCIS.**

**Keywords** Extracellular Contractile Injection System; eCIS; Microbial Toxins; Signal Peptide; Secretion Systems
**Subject Categories** Computational Biology; Microbiology, Virology & Host Pathogen Interaction

## Introduction

Bacteria engage in interactions with various hosts and microbes by employing proteins that are either secreted into the surrounding environment or directly injected into target cells via microbial multi-protein complexes known as secretion systems. These secretion systems can be classified into different types based on their molecular structures. Effectors, the secreted proteins, frequently play a crucial role as virulence factors that can impact eukaryotic cells, leading to pathogenesis and even mortality. Other effectors target bacterial cells and allow the effector-secreting microbes to outcompete neighboring bacteria.

While several secretion systems, such as the Type III, VI, and VI secretion systems (T3SS, T4SS, and T6SS), have been extensively investigated (Enninga & Rosenshine, 2009; Pinaud et al, 2018; Christie and Cascales, 2005; Terradot and Waksman, 2011; Sana et al, 2017; Allsopp and Bernal, 2023; Filloux et al, 2008), the extracellular contractile injection system (eCIS) remains a relatively poorly understood microbial secretion system. The eCIS is a protein complex that likely evolved from a tailed phage which lost its capsid (Sarris et al, 2014; Heiman et al, 2023). Instead of phage DNA, eCIS delivers a diverse range of protein toxins into target organisms (Heymann et al, 2013; Yang et al, 2006; Vlisidou et al, 2019; Rocchi et al, 2019; Geller et al, 2021; Casu et al, 2023; Wang et al, 2022). eCIS is a 110–120 nm and ~10 megadalton cell-free nano-syringe that is encoded in bacteria and archaea from multiple phyla (Geller et al, 2021; Sarris et al, 2014; Chen et al, 2019). Unlike most secretion systems which are membrane-bound, eCIS is unique by being extracellular following its release from the producing cell likely via lysis (Shikuma et al, 2014). Interestingly, eCIS is predominantly found in environmental microbes isolated from soil, water, invertebrates, and plants (Geller et al, 2021), but its specific roles in these environments remain largely unknown (Xu et al, 2022). However, different types of eCIS and intracellular CIS (contractile injection systems) are known to engage in a wide range of interactions, including those with invertebrates and fungi, as well as mediating programmed cell death and invertebrate developmental processes (Vladimirov et al, 2023; Heymann et al, 2013; Hurst et al, 2007; Yang et al, 2006; Shikuma et al, 2014; Nagakubo et al, 2021; Casu et al, 2023). The diverse activities associated with eCIS suggest that it may play a fascinating multifaceted role in the biology of the diverse microbes that encode it.

Among the interactions facilitated by eCIS, three types have been identified as having an impact on invertebrates. The

[1]Department of Plant Pathology and Microbiology, Institute of Environmental Sciences, The Robert H. Smith Faculty of Agriculture, Food & Environment, The Hebrew University of Jerusalem, Rehovot 7610001, Israel. [2]Department of Microbiology, University of Illinois Urbana-Champaign, 601 South Goodwin Ave, Urbana 61801 IL, USA. [3]Department of Entomology, Institute of Environmental Sciences, The Robert H. Smith Faculty of Agriculture, Food & Environment, The Hebrew University of Jerusalem, Rehovot 7610001, Israel. [4]School of Computer Science and Engineering, The Hebrew University of Jerusalem, Jerusalem, Israel. [5]Department of Developmental Biology and Cancer Research, Faculty of Medicine, The Hebrew University of Jerusalem, Jerusalem, Israel. [6]These authors contributed equally: Aleks Danov, Inbal Pollin. ✉E-mail: alevy@mail.huji.ac.il

Antifeeding prophage (Afp), produced by the entomopathogenic bacterium *Serratia entomophila*, causes feeding cessation and death to the New Zealand grass grub pest (Hurst et al, 2004). Another strain of *Serratia proteomaculans* produces eCIS particles named AfpX, which exhibit more rapid lethality compared to other Afp-producing strains (Hurst et al, 2021, 2018). The Photorhabdus virulence cassette (PVC) eCIS particles, produced by the *Photorhabdus* strains, have been shown to induce mortality when injected into *Galleria mellonella* larvae (Yang et al, 2006). Finally, the Metamorphosis-associated Contractile structures (MACs) are crucial for the development of the tubeworm *Hydroides elegans* (Shikuma et al, 2014). The intracellular CISs from *Streptomyces*, which are genetically related to eCIS, are implicated in regulated cell death in response to stress (Casu et al, 2023; Vladimirov et al, 2023; Nagakubo et al, 2023).

The typical structure of an eCIS operon has remained relatively conserved throughout evolution. The operon consists of core genes that encode structural proteins responsible for constructing the phage tail-like eCIS structure, including a contractile sheath, a baseplate with protruding tail fibers, a needle complex, and an inner tail tube that ends with a spike. Furthermore, the operon may also include the genes encoding eCIS-associated toxins (EATs) which are delivered via eCIS (Geller et al, 2021; Yang et al, 2006; Hurst et al, 2018; Xu et al, 2022; Vlisidou et al, 2019; Wang et al, 2022). In most cases, these EATs, often named effectors, are located a few genes upstream or downstream of the operon in the bacterial genome (Geller et al, 2021; Yang et al, 2006; Hurst et al, 2018; Xu et al, 2022; Vlisidou et al, 2019). Currently, only ~20 EATs have been identified and validated in laboratory settings, demonstrating broad effects on various types of cells and organisms, including invertebrates, mammalian and insect cell lines, as well as model organisms like *E. coli* and *S. cerevisiae* (Yang et al, 2006; Rocchi et al, 2019; Geller et al, 2021; Jank et al, 2015; Xu et al, 2022). Since most effectors serve as toxins, we will use the term EATs in this manuscript.

Despite the progress achieved in eCIS research, much remains unknown regarding the specific EATs that eCIS injects into target cells. Various methods have been explored for identifying EATs. Traditional bioinformatics approaches based on protein sequence conservation are frequently inadequate since EATs are not necessarily conserved (Geller et al, 2021). We and others previously identified that different eCIS operons, even when similar to each other, tend to encode distinct sets of EATs (Hurst et al, 2018; Yang et al, 2006; Geller et al, 2021). The EAT diversification process may result from the strong selective pressure imposed by EATs on the target cell, leading to evolved resistance by target hosts or microbes. Taking into account that eCIS loci are usually accompanied by EAT genes, that there are 1425 known eCIS loci, and that there are only ~20 known EATs, it is very likely that the vast majority of EATs await discovery (Geller et al, 2021). Consequently, new efficient approaches are needed to reveal EATs to understand the ecological role of eCIS as well as to discover new useful toxins. The rapid progress in the engineering of eCIS as a programmable target-cell specific protein delivery system (Kreitz et al, 2023; Jiang et al, 2022) combined with natural EATs that evolved for efficient eCIS-dependent delivery can open exciting opportunities in medicine, biotechnology, and agriculture.

Machine-learning approaches have been previously used for the classification of novel toxin genes and/or effectors associated with different secretion systems, but not specifically for EATs (Wang et al, 2018, 2019; Burstein et al, 2009, 2015; Sperschneider et al, 2016). Here, we present an XGBoost classifier that combines various genetic and biochemical features to provide a classification tool for EATs with ROC Area Under the Curve = 0.99. In addition, we could identify features that define a strong eCIS signal peptide at the EAT N-terminal sequence. Using our classifier, we classified 2194 genes as putative EAT genes from 950 eCIS-encoding genomes. Subsequently, we selected and tested EAT candidates to confirm their ability to kill prokaryotic or eukaryotic cells. We thereby discovered four new EATs (named EAT14-17) likely secreted by eCIS of four different microbes that were toxic to *E. coli*, *S. cerevisiae*, or both. An outline of the process is presented in Fig. 1. For EAT14-16, we identified residues that are critical for EAT activity, and their replacement with alanines reduced toxicity. Finally, we identified that EAT14 inhibits Rac1/Cdc42-mediated mitogenic signaling in human cells. To conclude, our study uncovers novel EATs and characterizes some of their functionalities, and provides thousands of predictions to be tested by many other labs.

# Results

## Data collection

We required negative (non-EATs) and positive (EATs) sets for the classifier development. To assemble the dataset, we compiled a negative set from the Integrated Microbial Genomes (IMG) database (Chen et al, 2020), which consisted of eCIS core genes, defined as those encoding structural secretion system components (Geller et al, 2021), random non-toxin genes, and effector genes from various non-eCIS secretion systems (T1SS-T4SS) (Materials and Methods). The positive set included 22 validated EATs, as well as high-likelihood EATs based on their genomic location within eCIS operons and the presence of known PFAM toxin domains. To reduce redundancy genes having over 90% similarity to each other were excluded, resulting in 146 genes for the positive set and 2949 genes for the negative set (Dataset EV1). For each gene/protein in the collected data, we extracted 11 features, as described below.

To evaluate the suitability of the extracted features for EAT classification, we assessed their ability to differentiate between EATs and non-EATs. An informative feature should exhibit a clear separation between the two groups. We therefore compared the distribution of each feature among the positive and negative samples in our dataset.

## Genomic features characterizing EATs

We observed that all genomic features showed significant separation between the two groups. EATs are more likely to be encoded in the eCIS operon vicinity: 88% of EATs are located up to ten genes away from the eCIS operon, compared to 1% of the non-EAT group (Fig. 2A). The DUF4157 is a clear signature domain of EAT proteins: 36% and 0.06% of EATs and non-EATs encode this domain, respectively (Fig. 2B). The DUF4157 domain tends to be encoded by up to two genes upstream/downstream to the EAT gene: 21% and 3% of EATs and non-EATs have this feature, respectively (Fig. 2C). In our previous work, we defined 'cloud eCIS

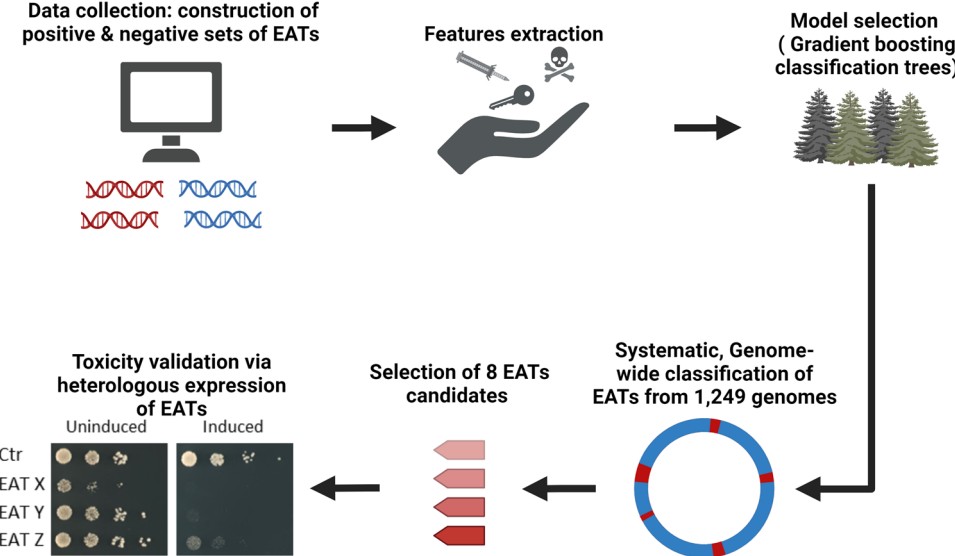

**Figure 1.  Machine learning-aided pipeline for EAT classification.**

Positive (EATs) and negative (non-EAT) genes were collected from the Integrated Microbial Genomes (IMG) database. Different genomic and biochemical features were extracted and used to train a gradient-boosting XGBoost classifier, using a fivefold cross-validation. Next, we performed a systematic genome-wide classification of all the genes from 1249 eCIS-encoding ("eCIS+") genomes. We selected EATs candidates and experimentally tested them for toxicity using heterologous expression in bacteria and yeast cells (drop assay images are re-used in Fig. 6).

domains' as domains that are enriched in eCIS genomic neighbor-hoods but are found only within eCIS operons from up to three phyla (Geller et al, 2021), unlike the more conserved eCIS core domains (>10 phyla) and shell domains (4–10 phyla). We observed that cloud domains are also characterizing EAT proteins: 56% and 12% of EATs and non-EATs have these domains, respectively (Fig. 2D).

## Biochemical features characterizing EATs and their usage in the prediction of eCIS signal peptide

We tested if different biochemical properties could be used for EAT classification: hydrophobicity, length, charge, aromaticity, and the presence of disordered regions in different subsequences of the protein (Materials and Methods). We also tested in which amino acid residue (aa) range we get the best separation between the EATs and non-EATs. The histograms of N-terminal hydrophobicity values, molecular weight, charge, disorder-promoting amino acids, and aromaticity for the EAT and non-EAT sets indicated potential discriminatory values between the two groups (Fig. 3). We found that the first 35 residues of EATs exhibited higher hydrophilicity compared to non-EATs (Fig. 3A, KS test, $P = 7.06e-54$). Further-more, the molecular weight of the first eight residues was higher for EATs vs. non-EATs (Fig. 3B, KS test, $P = 4.11e-11$). The charge of residues 6–21 was higher for EATs vs. non-EATs (Fig. 3C, KS test, $P = 1.44e-14$), and residues 6–21 of EATs had a significantly higher number of disorder-promoting amino acids than non-EATs (Fig. 3D, KS test, $P = 1.72e-34$). The first eight residues had a higher propensity to include at least one aromatic acid for EATs and non-EATs (Fig. 3E, KS test, $P = 1.06e-9$).

Importantly, the N-terminal end of EATs should carry the eCIS signal peptide, which was shown to be required for effector loading

and translocation of the PVC and Afp eCISs, but a specific conserved sequence was not detected (Jiang et al, 2022; Kreitz et al, 2023; Steiner-Rebrova et al, 2023). Therefore, it is likely that the N-terminal signal peptide is being recognized through its folded peptide structure. The discriminatory power of these N-terminal features suggests that the eCIS signal peptide may carry distinguishable biochemical properties. Indeed, the experimentally confirmed eCIS signal peptide of Pdp1 (Jiang et al, 2022; Kreitz et al, 2023), which was part of our positive set (Dataset EV1, IMG gene ID 644889759), has high hydrophilicity (hydrophobicity = −36.9 on the Kyte–Doolittle hydrophobicity scale), a relatively high molecular weight (1158 dalton for the first 10 aa), a positive N-terminal charge (4), and two aromatic acids within the first eight aa (MPRYANYQ). Therefore, we posit that these biochemical features can be used to reliably identify eCIS signal peptides.

We set out to identify a combination of the features that could be used to describe an eCIS signal peptide score. We used three biochemical features that showed the best potential in separating the two groups - hydrophobicity, aromaticity, and disorder-promoting index (Fig. 3A,D,E). In order to find the optimal weighting of these features, we combined these features as independent variables to a support vector machine (SVM) classifier that categorized proteins as EAT or non-EAT. Based on this classifier and for each gene in EAT and non-EAT sets of proteins, we evaluated the probability that the protein belongs to the EATs class, which yielded an eCIS signal peptide score between 0–1 with ROC AUC of 0.91 (Fig. 4A). Based on the score distribution for EATs and non-EATs (Fig. 4B) we divided the score to weak (score ≤0.1), medium (0.1 < score ≤ 0.2), and strong signal peptide (score >0.2) for an intuitive interpretation. This eCIS signal peptide score separates well between known EATs and non-EATs (Fig. 4C). Moreover, we tested if the signal peptide score is independent of the eCIS genomic features (Fig. 4D–F). We observed a

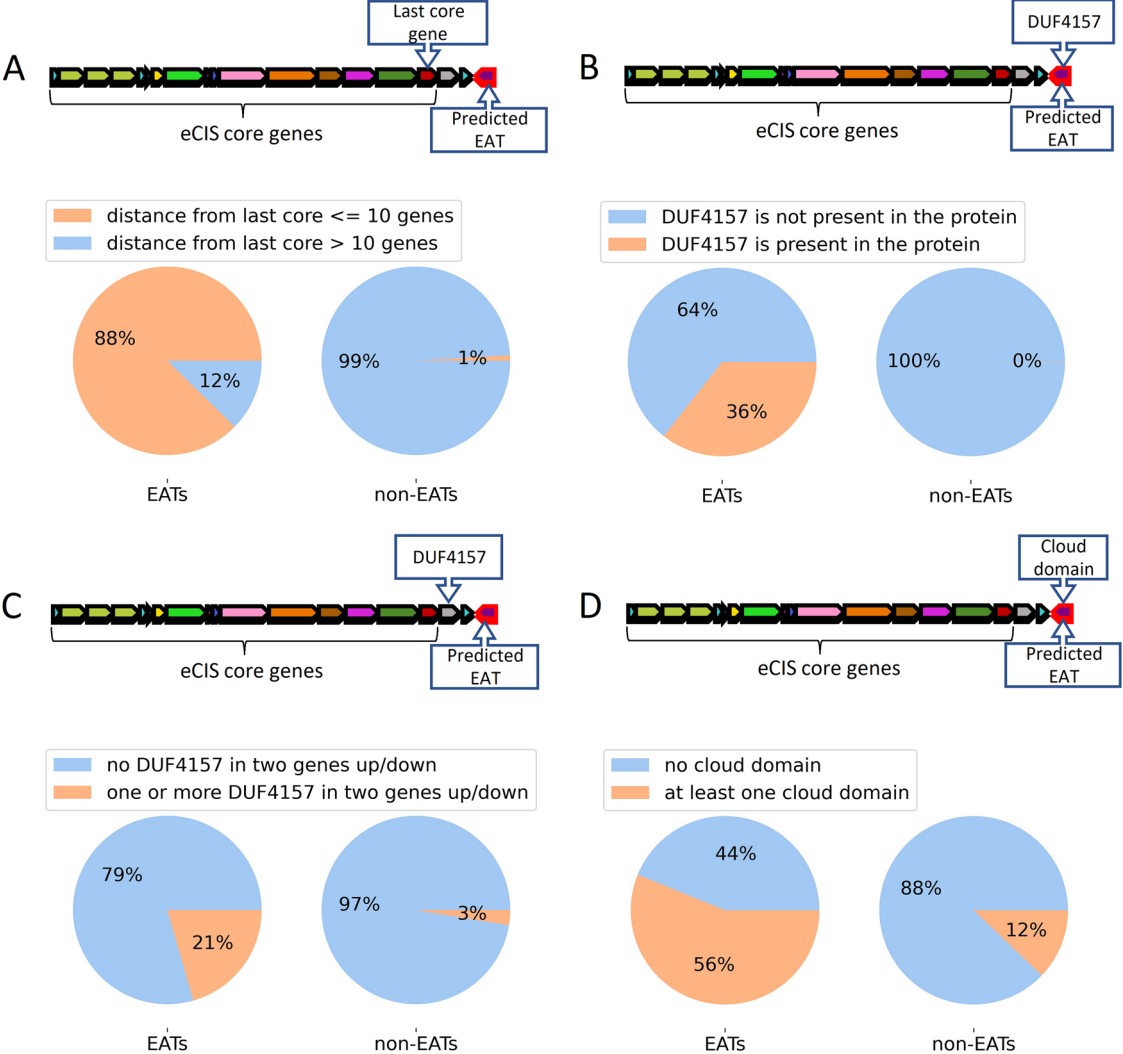

**Figure 2. The genomic features and their percentages among the EATs (*n* = 146) and the non-EAT genes (*n* = 2949).**

The different colors denote different eCIS core genes within the operon. (**A**) Distance from the last core feature distribution (without core genes). The EAT genes are almost entirely located less than ten genes away from the last core gene, while the background group has the opposite distribution. (**B**) DUF4157 domain presence feature distribution. (**C**) DUF4157 domain gene adjacency feature distribution. (**D**) Cloud domains feature distribution.

strong correlation between the presence of the DUF4157 domain and having a strong signal peptide score (Fig. 4D), suggesting that these are two independent features that together point to a reliable EAT prediction. Out of the proteins with DUF4157 domains, 61% have a strong signal peptide, whereas out of the proteins without DUF4157 domains, only 5% have a strong signal peptide. Other features, the presence of DUF4157 in neighboring genes and the presence of cloud domains, are also correlated with strong signal peptides but to a lesser extent than the presence of DUF4157 (Fig. 4E,F).

## Development of eCIS-associated toxin automated classifier algorithm

To classify the potential EATs, we developed a gradient-boosting XGBoost Random Forest classifier, which combines an ensemble of decision trees. XGBoost Random Forest classifier is known to be

robust to imbalanced datasets, where one class (e.g., negative samples in our case) is more abundant in the training data compared to the other class (More and Rana, 2017). Here, we trained an ensemble model, composed of 40 decision trees with a maximum depth of seven, and a minimum leaf size (the minimum number of samples in the classification node of the tree) of four. These parameters yielded the best evaluation metrics in terms of positive predictive value (PPV). The performance metrics of the trained XGBoost model, using the chosen parameters, were assessed through fivefold cross-validation. The inputs included all the genomic and biochemical features described above. The results demonstrated an average PPV of 0.94, indicating that 94% of the samples classified as EATs by the algorithm were true effectors. Moreover, the algorithm successfully classified 85% of all positive samples in the test set as EATs. We also tested our model in terms of the receiver operating characteristic curve (ROC) curve and

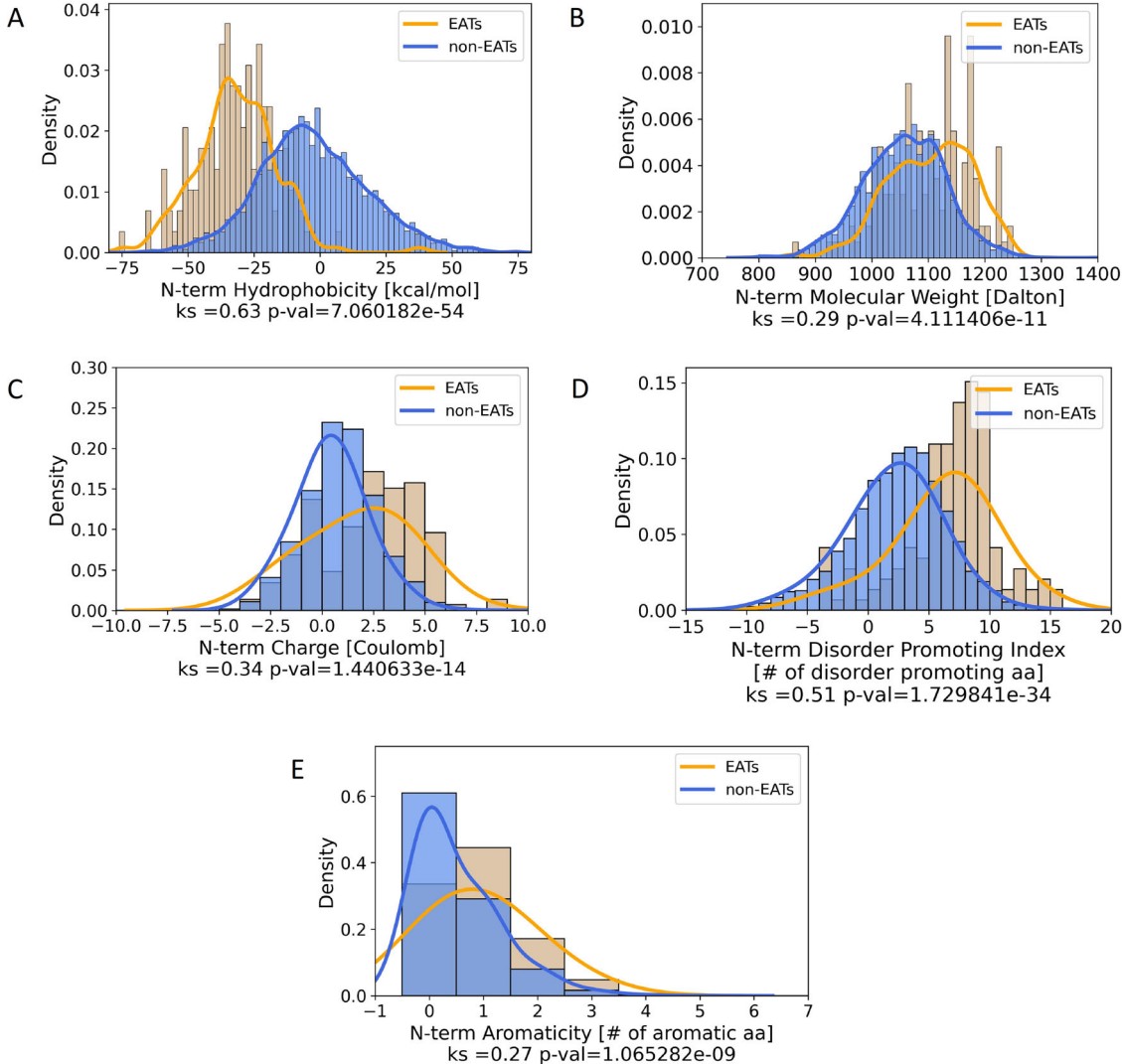

**Figure 3. The distribution of the most significant biochemical (sequence-based) features along segments of the protein N-terminus.**

(A) Histogram of first 35 N-terminal amino acids mean hydrophobicity according to the Kyte–Doolittle scale. Namely, the N-termini of EATs are more hydrophilic. (B) Histogram of first eight N-terminal amino acids mean molecular weight. (C) Histogram of 6–21 n-terminal amino acids mean charge. (D) Histogram of 6–21 N-terminal amino acids mean disorder-promoting index. (E) Histogram of the number of aromatic amino acids in eight first N-terminal residues.

received ROC area under the curve (AUC) of 99% (Fig. 5A). These high metrics suggest the precision and robustness of our approach for EAT classification, instilling confidence in its ability to discover novel EATs.

Furthermore, we computed the SHapley Additive exPlanations (SHAP) values for each feature to elucidate the model's behavior in relation to the individual features' incremental impacts on model performance. Briefly, SHAP values are game theory-based scores that quantify how important each feature was when the label of each sample was predicted. Notably, the distance from the last core, N-terminal hydrophobicity, N-terminal order, and the presence of cloud domains emerged as the most influential features (high absolute SHAP values in Fig. 5B). Conversely, the presence of the prokaryotic canonical signal peptide, which is used to deliver proteins via SecYEG and is unrelated to eCIS, was observed to be of no significance. Intriguingly, it was evident that positive values of

features linked to DUF4157 exerted a strong favorable influence on the model's affirmative predictions, whereas negative values of these features showed limited contributions to the algorithm's predictions, suggesting that DUF4157 domain is likely sufficient for positive eCIS prediction, but its absence is not necessarily informative (Fig. 5B). The success of the XGBoost model was based on both groups of genomic and biochemical features, highlighting the benefit of combining them into one model.

Next, we extracted all 6,867,673 protein-coding genes from 1249 eCIS operon-positive genomes ("eCIS+ genomes") from our previous study (Geller et al, 2021). We applied the classifier to this dataset and each gene received a score ranging between 0 (non-EAT) to 1 (high confidence EAT). Of these, the trained model classified 2194 genes from 950 genomes as potential EATs, having a prediction score higher than 0.5 (Dataset EV4). We also added the eCIS signal peptide score to our predictions to allow researchers to

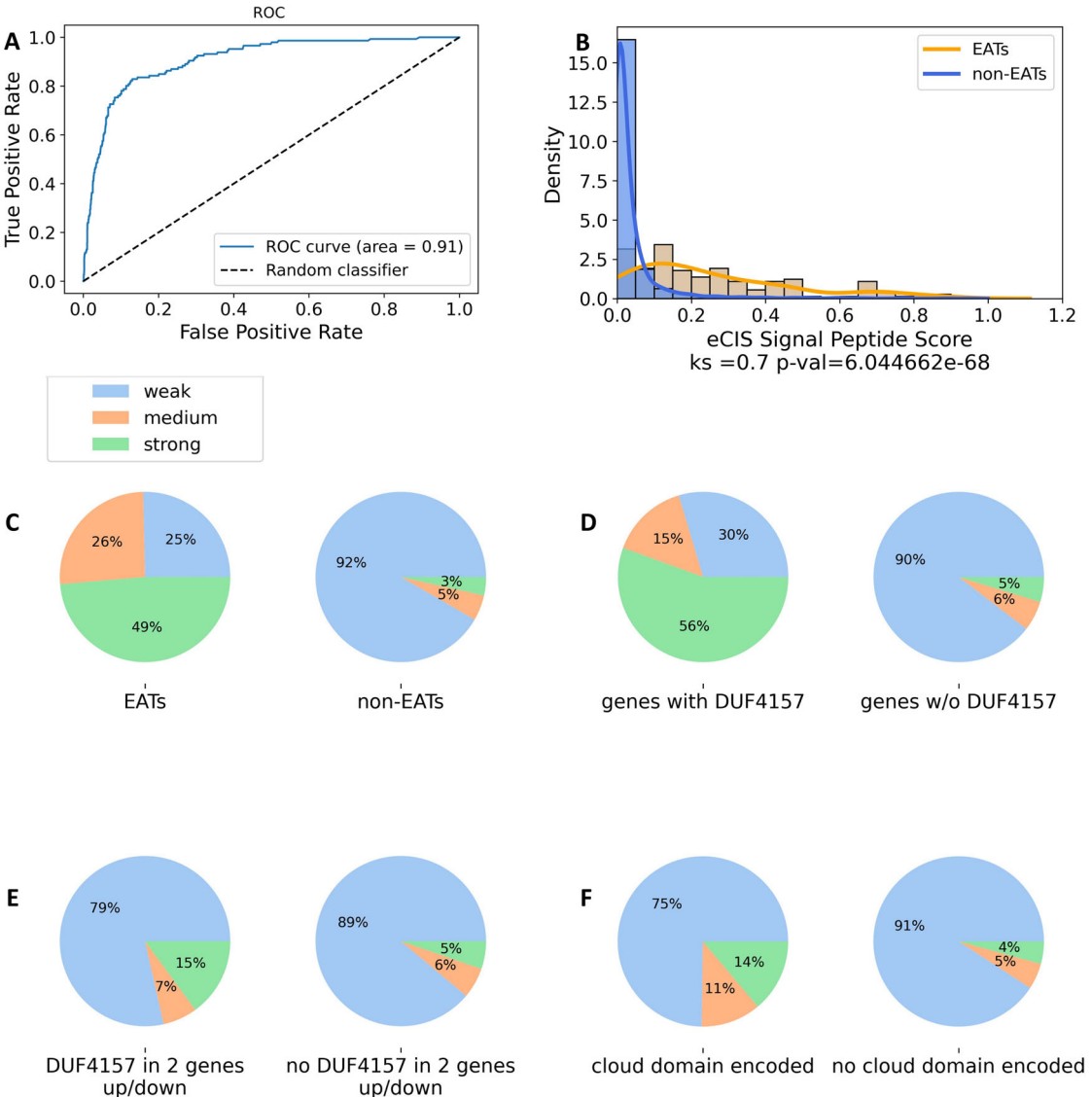

**Figure 4. The signal peptide SVM ROC and the classification of EATs and non-EATs and associated features based on the new eCIS signal peptide score.**

(A) The receiver operating characteristic (ROC) curve of the SVM model trained upon the three sequence-based features (n-term hydrophobicity, n-term disorder-promoting index, and aromaticity) with the area under the curve (AUC) = 0.91. (B) The distribution of score values for positive and negative sets. (C) The percentage of the score values in EATs and non-EATs. The weak signal peptide score characterizes 92% of the non-EATs group. Conversely, within the EAT group, 75% of the sequences have either medium or strong signal peptide scores. (D) The percentage of the score quality within the proteins in our dataset that encode the DUF4157 domain within their protein sequence and the group of genes that do not encode DUF4157. (E) The percentage of the score within the group of genes that are adjacent to DUF4157 (two genes up/downstream in the genome) and those that are not adjacent to DUF4157. (F) The percentage of the score in the group of genes that encode one or more cloud domains and in those that do not encode these domains.

use strong native signal peptide sequences to allow an efficient eCIS loading of EATs for different applications. The average score is 0.249, which is in the range of strong signal peptides (>0.2). The top and bottom 100 EAT candidates sorted by EAT prediction score (0.5–1.0) have an average signal peptide score of 0.405 and 0.225, respectively.

We analyzed the distribution of predicted EATs per genome and genus. Some organisms were rich in predicted EATs suggesting a strong involvement of eCIS in their physiology and ecology (Fig. EV2). For example, the aquatic filamentous Cyanobacteria *Moorena producens* PAL 15AUG08-1 (IMG genome ID

2630968268), also called *Moorea*, contains 18 predicted EATs, half of which include a DUF4157 domain, and the other half include eCIS cloud domains such as XisH and XisI. We previously confirmed the toxicity of EAT9, a restriction endonuclease, encoded by this specific genome (Geller et al, 2021). The *Moorea* genus, in general, is rich in predicted EATs, with an average of 7.8 EATs per genome in the six *Moorea* eCIS+ genomes (Fig. EV2). *Moorena producens* is implicated in the poisoning of humans and green turtles and a future study could test whether eCIS is associated with this phenotype (Curren et al, 2022). Other examples of EAT-rich microbes include the marine microbe *Fulvivirga*

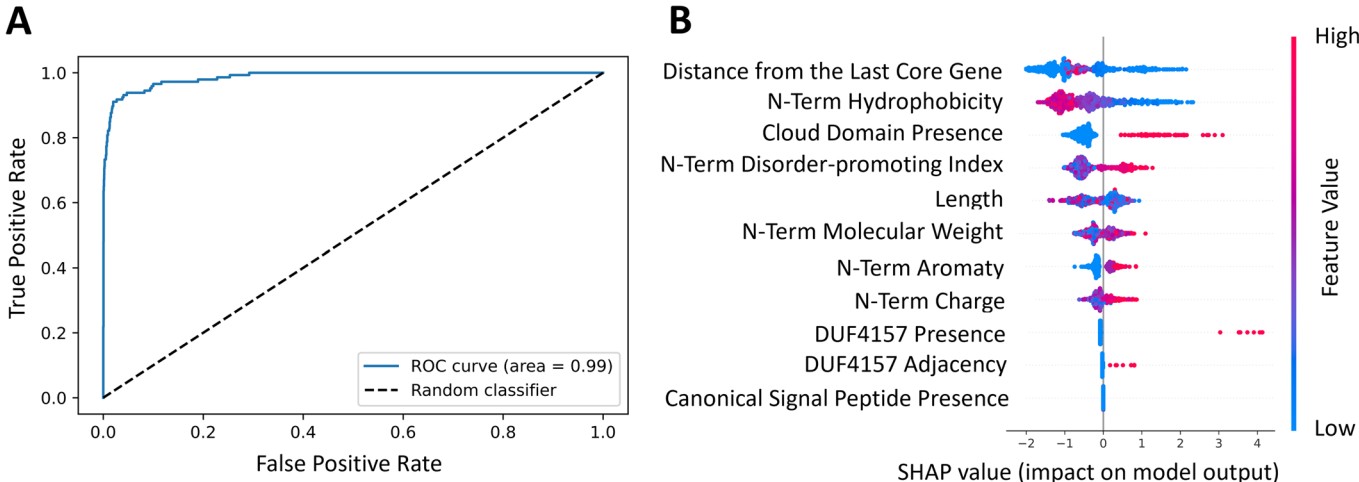

**Figure 5. Performance of the EAT XGBoost classification model and its features.**

(A) The ROC of the model. The curve shows the performance of five different models trained on 80% of the data and tested on 20% of the data during a fivefold cross-validation process, for different classification thresholds. (B) The impact of the different features on the model outcome. A beeswarm plot of the SHapley Additive exPlanations (SHAP) values. Each dot represents one sample from the feature matrix. The values are grouped by features and colored by feature values. The red points are high values of the feature and the blue points are low values. The x-axis represents the SHAP value. Accordingly, the distance from the last core gene and the N-term hydrophobicity have the highest contribution to the model.

*imtechensis* AK7 and the soil bacteria *Cylindrospermum stagnale* PCC 7417 which encode 16 and 13 predicted EATs, respectively. In contrast, the insecticidal *Photorhabdus* genus, which is extensively studied for its PVC eCIS, and has 3–5 eCIS operons per genome (Geller et al, 2021), has on average 3.8 predicted EATs/genome (Figure EV2), including the confirmed Pnf (Yang et al, 2006; Vlisidou et al, 2019), Pdp1 (Wang et al, 2022), RRSP (Wang et al, 2020b), and EAT16 from this work as explained below. Among the archaea, the methylotrophic methanogen *Methanomethylovorans hollandica* DSM 15978 has the most predicted EATs/genome with five EATs.

We analyzed the functionality of the predicted EATs. All predicted EATs had at least one Pfam domain. Out of the predicted EATs, 1130 (51.5%) contained a DUF4157 domain, most of them (*n* = 983) without an additional annotated domain which might not have been identified yet. Indeed, several years ago, a novel DNAse toxin domain was identified to be fused to DUF4157 (Jana et al, 2020). In addition, 69 proteins (3.1%) were annotated as nucleases or RNAses, 46 proteins (2%) were annotated as proteases or peptidases, and 32 proteins (1.4%) were annotated as other toxins, such as BrnT, HigB-like, Cif, and YoeB toxins. Importantly, these results indicate that a vast majority of EATs encode novel types of toxins with potentially new modes of action for cytotoxicity.

## Four new EATs kill bacterial and yeast cells

To assess the cytotoxicity of predicted EATs, eight candidates with high prediction scores (>0.93, Dataset EV4) were heterologously expressed on plasmid vectors in *E. coli* and/or *S. cerevisiae* (Table 1). We tested each prediction with or without induction to quantify toxicity. Induction in *E. coli* and *S. cerevisiae* was done by adding 0.2% arabinose and 2% galactose to the media, respectively,

whereas repression in *E. coli* and *S. cerevisiae* was done by adding 1% glucose and 2% glucose to the media, respectively.

We decided to test certain predicted EAT genes in a specific target organism based on the taxonomic distribution of the predicted toxin domains as described on the Interpro website (Paysan-Lafosse et al, 2023). Namely, proteins with domains that mostly appear in bacteria were tested in *E. coli* and domains that appear in many eukaryotes were tested in *S. cerevisiae*. For example, domain PLA2G12 from the ML_1 gene is mostly a eukaryotic domain (96% of its occurrences are in eukaryotes), and therefore we thought that the protein containing it might function as a toxin that targets eukaryotes that was acquired by bacteria for this purpose. Overall, three candidate genes were tested in *E. coli*, three were tested in *S. cerevisiae* and two were tested in both organisms. We did not test all combinations due to resource constraints.

Among the eight candidates, four demonstrated significant toxicity against bacteria, yeast, or both, leading to at least a 100-fold reduction in colony-forming units (cfu, Fig. 6A,B, full results are in Fig. EV3). All four effectors were located downstream of their cognate eCIS operons (Fig. 6C). Using a similar approach, we previously confirmed 13 new EATs, EAT1-13 (Geller et al, 2021). Therefore, we named the newly confirmed toxins EATs 14-17. EAT14 and EAT15 exhibited cytotoxicity against both *E. coli* and *S. cerevisiae*, suggesting a conserved activity for these toxins, similar to what we have previously observed for EAT1, EAT3, and EAT11 (Geller et al, 2021).

EAT14 from the soil microbe *Chitinophaga rupis* was the most toxic protein in our assays, leading to a ~10,000-fold reduction in cfu/mL when expressed in *E. coli* compared to uninduced EAT14 (Fig. 6A, compare left vs. right panel) and was also highly toxic to *S. cerevisiae* even without expression induction (Fig. 6B left). EAT14 is characterized by the presence of an N-terminal DUF4157 domain and contains a transglutaminase-like domain at its C-terminus

**Table 1. List of EAT candidates tested and their cytotoxicity.**

| Gene name | Tested organism | Organism name | IMG gene ID | ML score | Predicted function based on structure or Pfam domain | Confirmed cytotoxicity to E. coli? | Confirmed cytotoxicity to S. cerevisiae? |
|---|---|---|---|---|---|---|---|
| ML_1 | S. cerevisiae | Halogeometricum rufum CGMCC | 2618035200 | 0.93 | DUF4157; PLA2G12 (PF06951); LysM | Not tested | - |
| ML_2 (EAT14) | E. coli & S. cerevisiae | Chitinophaga rupis DSM 21039 | 2623476966 | 0.93 | DUF4157; LysM; Transglut_prok (PF09017) | + | + |
| ML_3 (EAT15) | E. coli & S. cerevisiae | Aquimarina atlantica | 2578564370 | 0.95 | TIR domain | + | + |
| ML_4 (EAT17) | S. cerevisiae | Mycetohabitans rhizoxinica HKI 454 | 650722981 | 0.99 | PGAP1 (PF07819); Hydrolase alpha/beta | Not tested | + |
| ML_5 | E. coli | Morphospecies DK DK162 | 2727705283 | 0.99 | PF13711 | - | Not tested |
| ML_6 (EAT16) | E. coli | Photorhabdus aegyptia BA1 | 2557154383 | 0.94 | DUF4365; Holliday Junction Resolving Enzyme | + | Not tested |
| ML_7 | S. cerevisiae | Aquimarina sp. AU58 | 2606519444 | 0.98 | DUF4157; DUF4573 | Not tested | - |
| ML_8 | E. coli | Geobacter sp. M21 | 644869287 | 0.99 | DUF599; restriction endonuclease | - | Not tested |

(Fig. 7A). Transglutaminase (TG) post-translational modification of target proteins has been demonstrated for microbial toxins like the cytotoxic necrotizing factor 1 (CNF1) from *E. coli* and the dermonecrotic toxin (DNT) from *Bordetella* (Buetow et al, 2001; Fukui and Horiguchi, 2004). TG proteins from prokaryotes and eukaryotes contain a catalytic Cys-His-Asp triad or a Cys-His dyad (Hashizume et al, 2011; Noguchi et al, 2001; Wilson and Ho, 2010; Yu et al, 2015; Steffen et al, 2017; Fernandes et al, 2015). Based on structural modeling using AF2, we predict that the Cys-His-Asp triad might also constitute the active site of EAT14 (Fig. 7B). To assess the significance of this triad for EAT14, we generated and tested independent point mutations at positions C298A, D397A, and H410A. As expected, all three mutations led to the loss of cytotoxic activity (Fig. 7C). Notably, several TG-like proteins have been identified in the past as proteases (Sanchez-Pulido and Ponting, 2016; Ozhelvaci and Steczkiewicz, 2023; Makarova et al, 1999), hence we cannot rule out at the moment the possibility that EAT14 actually serves as a protease and not as a TG.

EAT15, which was also highly toxic to bacteria and yeast, possesses a TIR domain in its N-terminus (Fig. 7D). TIR domains of bacteria and archaea have NADase activity which cleaves the essential molecule NAD+ (Essuman et al, 2018). TIR domain also confers cytotoxic activity against yeast, plants, and mammalian cells, indicating the broad activity of the TIR domain (Coronas-Serna et al, 2020; Eastman et al, 2022; Essuman et al, 2018). We previously identified that EAT5 also possesses a NADase activity via a RES domain (Geller et al, 2021). We compared EAT15 predicted structure against PDB proteins using foldseek (van Kempen et al, 2023) and identified a strong hit against bacterial TIR domain from *Paracoccus denitrificans* (PDB ID 3H16) (Chan et al, 2009) (Fig. 7D). We observed a predicted active site at position E94 of 3H16 that corresponds to position E83 of EAT15 (Fig. 7E). We replaced E83 with Alanine and the point mutation led to a loss of activity, as illustrated in Fig. 7F.

EAT16 comprises two distinct domains based on its folding using Alphafold2. Its N-terminus contains a DUF4365 and bears similarity to Holliday Junction resolvase (HJR) (Fig. 7G). HJR is a critical enzyme involved in DNA recombination (Aravind et al, 2000), suggesting that EAT16 may function as a nuclease or as a DNA-binding protein, primarily targeting DNA. DUF4365 was recently shown to act as a DNAase leading to a nonspecific DNA degradation (Lu et al, 2024), suggesting a similar mode of action to EAT16. According to InterPro databases, the DUF4365 Hidden Markov Model (HMM) has a conserved peptide motif DxGxD… QxK. In EAT16, these residues were situated at positions D42, D46, Q66, and K68 (Fig. 7H). Notably, mutating three of the four conserved amino acids individually resulted in the loss of functional cytotoxicity (Fig. 7I).

EAT17 is from the fungal bacterial endosymbiont *Mycetohabitans rhizoxinica* HKI 454, which was isolated from the fungus *Rhizopus microsporus* (Partida-Martinez et al, 2007). In our assay, it was toxic to yeast which suggests antifungal properties from eCIS that injects EAT17. It contains annotation for post-glycosylphosphatidylinositol attachment to protein 1 (PGAP1)-like protein domain (PF07819). A toxin similar to EAT17 is the type VI secretion system (T6SS) effector TplE from *Pseudomonas aeruginosa*. It is a PGAP1-like phospholipase effector that inhibits bacterial growth and disrupts ER trafficking of mammalian cell lines (Jiang et al, 2016).

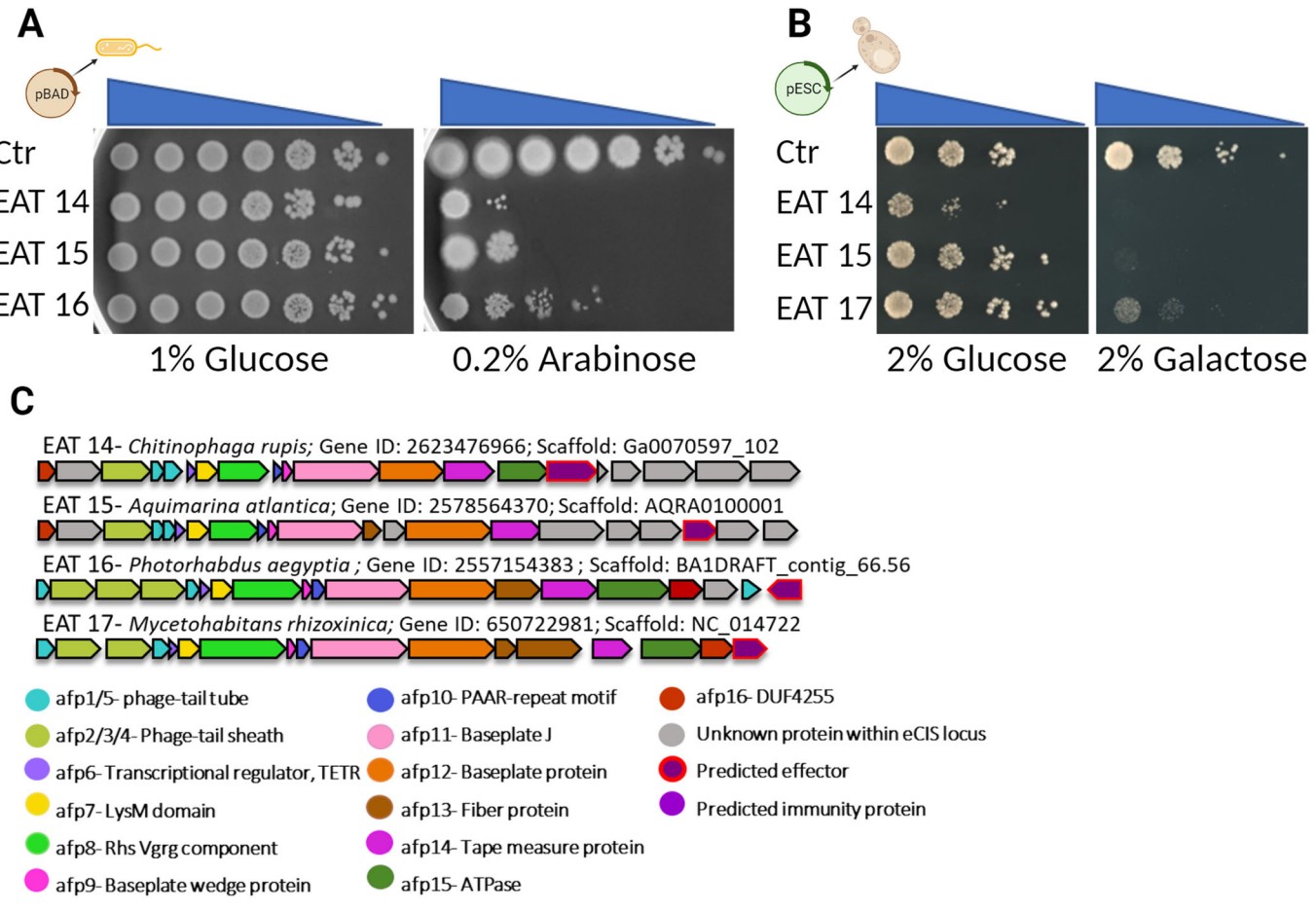

**Figure 6.  Cytotoxicity of the predicted EATs to *E. coli* and/or *S. cerevisiae*.**

(**A**) A representative drop assay of the EATs that were cytotoxic to *E. coli*. Ctr - bacteria with empty pBAD24 plasmids. Glucose leads to cloned gene repression and arabinose leads to its induction. Bacteria were serially diluted from left to right to quantify CFUs. (**B**) A representative drop assay of yeast EATs. Ctr - yeast cell with empty pESC plasmids. Glucose leads to cloned gene repression and galactose leads to its induction. Yeasts were serially diluted from left to right to quantify CFUs. (**C**) Schematic representation and annotation of each validated predicted EAT and its cognate eCIS operon. Source data are available online for this figure.

## EAT14 inhibits Rac1/Cdc42-mediated mitogenic signaling in human cells

We followed up experimentally on the possible mode of action of the new EATs. Many bacterial toxins are known to modulate mitogenic signaling pathways in mammalian cells via Rho family GTPases (RhoA, Rac1, Cdc42), including CNF1 and DNT (Chaoprasid and Dersch, 2021; Aktories, 1997; Popoff, 2014; Ho et al, 2018). Specifically, EAT14 shares the predicted TG activity of CNF1 and DNT (based on domain presence), and therefore we tested if it affects signaling pathways in eukaryotes like these well-established toxins. Previous studies have demonstrated that Rho GTPase signaling activates the mitogenic signaling pathway involving the transcription factor serum response factor (SRF) (Chai and Tarnawski, 2002) that binds to its cognate serum response element (SRE), which in turn regulates the expression of many genes controlling cell growth and differentiation and other mitogenic cellular functions (Treisman, 1992). Activation of the Rho-dependent SRF-SRE mitogenic signaling can be monitored by the use of a luciferase reporter gene fused to the SRE element

(Repella et al, 2013, 2011). Knowing that many toxic effectors act through GTPases involved in SRF-SRE-dependent mitogenic signaling, we tested the effect of EAT14-17 on RhoA-, Rac1-, and Cdc42-dependent mitogenic signaling in human embryonic kidney cells (HEK293T) by using the SRE-luciferase reporter gene assay, as previously described (Repella et al, 2013, 2011). We cloned and expressed the four EATs 14-17 within mammalian expression vectors and transfected them into HEK293T cells. The cells included two reporter gene plasmids: an SRE promoter fused to a firefly luciferase reporter gene and a herpes simplex virus TK promoter fused to a *Renilla* luciferase gene, as a low-expression constitutive reporter control gene. We measured the SRE promoter activity in response to the expression of the different EATs and a positive control VopC, a known activator of Rac1 and Cdc42 that stimulates SRE signaling (Zhang et al, 2012).

Compared to VopC, none of the EATs activated SRE-luciferase reporter activity in HEK293T cells (Fig. 8A); however, we did notice that EAT14, unlike the other EATs, consistently decreased the background SRE activity. We further evaluated the inhibitory effect of EAT14 on SRE-dependent signaling, by examining

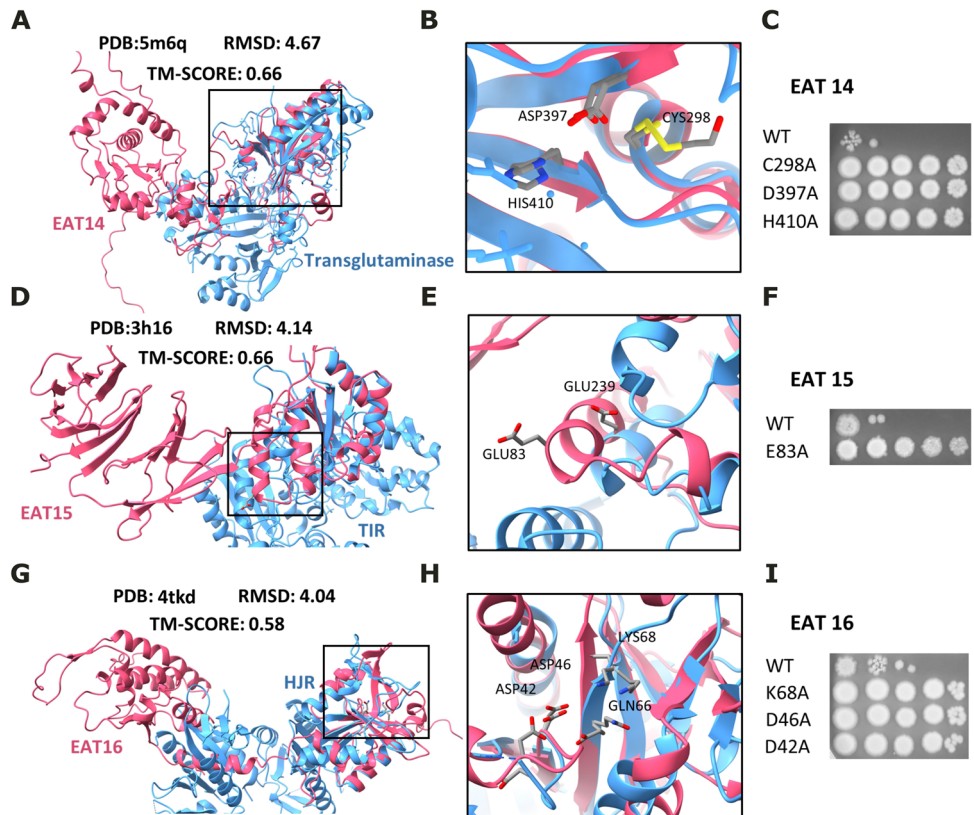

**Figure 7. Predicted protein structures and important residues of new bacterial EATs.**

(A) Structural similarity between EAT14 and known Transglutaminase struct. Magenta- the predicted EAT; Light blue- PDB accession 5m6q. Protein alignment is displayed using ChimeraX. (B) The predicted active sites of EAT14. Residue three-letter codes and position numbers are indicated. (C) Toxicity of the EAT14 and its mutants. The wild-type toxins (WT) and the EATs with mutated residues (changed to Ala) were cloned into *E. coli* BL21(DE3) under the arabinose-induced promoter and were tested for toxicity using a drop assay with serial dilutions to quantity CFUs. (D) Structural similarity between EAT15 and a protein with a TIR domain structure. Magenta- the predicted EAT; Light blue- PDB accession 3h16. The PDB accession of the target protein is presented. Protein alignment is displayed using ChimeraX. (E) The predicted active sites of EAT15. Residue three-letter codes and position numbers are indicated. (F) Toxicity of the EAT15 and its mutants. The wild-type toxins (WT) and the EATs with mutated residues (changed to Ala) were cloned into *E. coli* BL21(DE3) under the arabinose-induced promoter and were tested for toxicity using a drop assay with serial dilutions to quantity CFUs. (G) Structural similarity between EAT16 and a protein with an HJR domain structure (PDB accession 4tkd). Magenta- the predicted EAT; Light blue- PDB closest annotation based on foldseek structural search. The PDB accession of the target protein is presented. Protein alignment is displayed using ChimeraX. (H) The predicted active sites of EAT16. Residue three-letter codes and position numbers are indicated. (I) Toxicity of the EAT16 and its mutants. The wild-type toxins (WT) and the EATs with mutated residues (changed to Ala) were cloned into *E. coli* BL21(DE3) under the arabinose-induced promoter and were tested for toxicity using a drop assay with serial dilutions to quantity CFUs. Source data are available online for this figure.

whether EAT14 could inhibit VopC-dependent activation of SRE-luciferase activity. As shown in Fig. 8B, EAT14 interferes with VopC-activation of SRE signaling in a dose-dependent manner. This finding suggests that EAT14 activity occurs directly or indirectly through the Rac1 and/or Cdc42 signaling pathway.

## Discussion

The identification and characterization of novel EATs secreted by the extracellular contractile injection system (eCIS) is essential for comprehending the diverse roles played by this unique microbial secretion system (Vladimirov et al, 2023; Heymann et al, 2013; Hurst et al, 2007; Yang et al, 2006; Shikuma et al, 2014; Nagakubo et al, 2021; Casu et al, 2023; Desfosses et al, 2019). In this study, we employed for the first time a systematic approach to identify putative eCIS-associated toxins (EATs) across hundreds of

microbial genomes. We applied a machine learning approach using the XGBoost classifier to scan eCIS+ genomes and identify EATs based on eight genomic and biochemical features. By applying our classification tool to all eCIS+ genomes we identified 2194 putative EAT genes. Subsequently, we selected and tested eight EAT candidates for their ability to kill prokaryotic or eukaryotic cells in heterologous expression experiments, resulting in the discovery of four new EATs, EAT14-17. We identified critical residues that we predict to be localized within the EAT active sites as their mutagenesis abolished toxicity. We also developed a model for the eCIS N-terminal signal peptide which we believe can aid in efficient natural and artificial effector loading into eCIS particles.

Our study showcases the power of machine-learning algorithms in the identification and characterization of toxins, providing a valuable tool for prioritizing potential candidates for further experimental investigations. Machine-learning algorithms offer several advantages over traditional bioinformatics approaches for

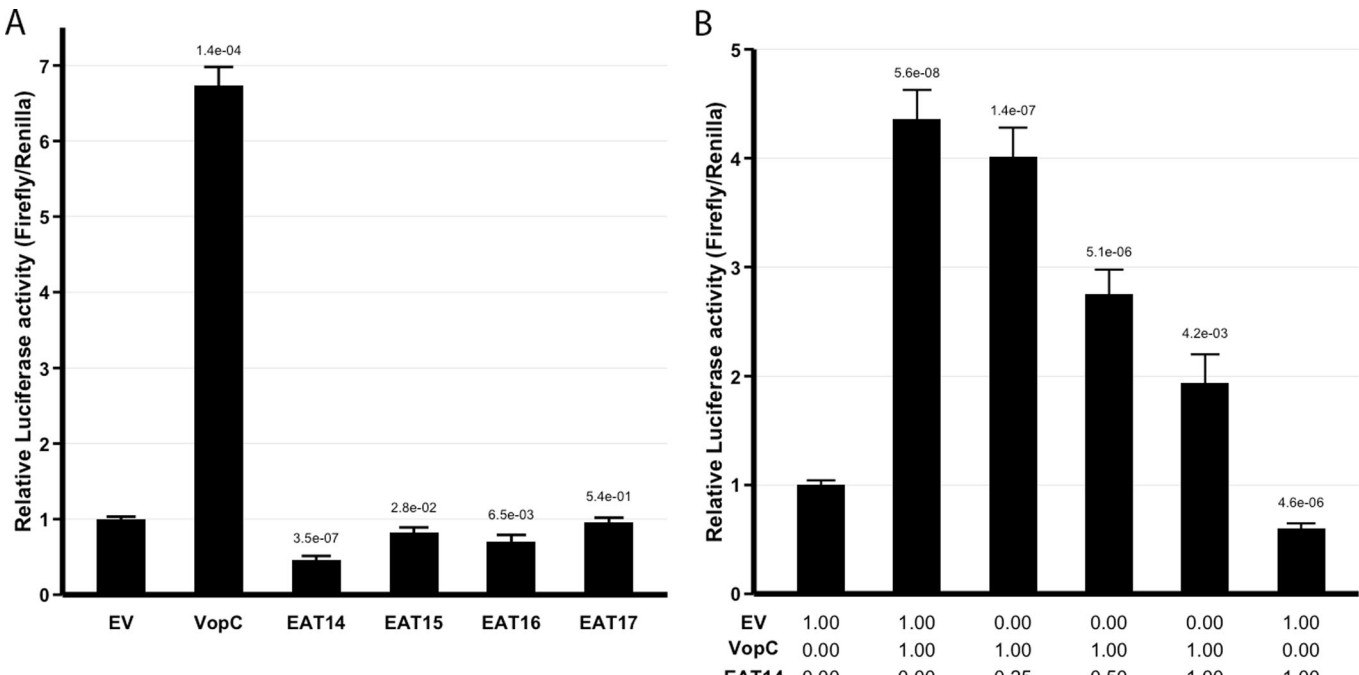

**Figure 8. EAT14 but not EAT15-17 inhibits SRE-dependent mitogenic signaling in HEK293T cells.**

Cellular SRE-luciferase reporter response in HEK293T cells cotransfected with mammalian expression plasmids encoding EAT14-EAT17, positive control VopC, or negative control empty vector (EV), and the dual-luciferase reporter plasmids, as described in Methods. Luciferase activity was determined by dividing the firefly RLUs by the Renilla control RLUs. The relative luciferase activity was determined by dividing the luciferase activity for experimental samples by the mean luciferase activity for the EV control samples of each repeat. All data points were performed in replicates of 4 and repeated at least three independent times. Data points shown are the mean ± stdev. (A) Comparison of the SRE-luciferase reporter response in HEK293T cells transfected with the indicated plasmids. (B) Cellular SRE-luciferase reporter response in HEK293T cells cotransfected with the ratios of VopC, EAT14, and EV plasmids used in the transfections indicated below the x-axis. Source data are available online for this figure.

gene classification based on protein sequence conservation (Wang et al, 2018, 2019; Burstein et al, 2009, 2015; Sperschneider et al, 2016). Many of the known EATs are species- or genus-specific, making it challenging to identify them solely based on sequence similarity. However, to achieve optimal performance, machine-learning algorithms necessitate robust data representations. Given the scarcity of positive data within the eCIS-associated toxins, we employed an augmentation technique to significantly expand our positive set by incorporating a substantial number of putative EATs. Additionally, we aimed to enhance the representation of the non-EATs genes (the negative set) by including different types of non-EATs genes - core genes, type I-IV effectors and genes randomly sampled from the eCIS+ genomes spanning from the eCIS scaffold and beyond.

Moreover, a conserved signal peptide sequence used by eCIS to load EATs was not detected, and the signal peptide activity was found to be resilient to mutations (Jiang et al, 2022; Steiner-Rebrova et al, 2023). An important outcome of our classifier is the future ability to design new EATs or non-toxic protein payloads, such as genome editor proteins, antimicrobial peptides, or other therapeutic proteins (Jiang et al, 2022; Kreitz et al, 2023; Steiner-Rebrova et al, 2023), by adding a custom N-terminal eCIS signal peptide and a DUF4157 domain upstream to a toxin domain. Based on our biochemical features an eCIS signal peptide should be very hydrophilic in its first 50 aa (Fig. 3A), include some relatively heavy and aromatic aa in the first eight aa (Fig. 3B,E), and aa 6–21 should

be slightly positively charged with a disordered structure (Fig. 3C,D). The predicted peptide is likely unstructured, thereby facilitating recognition of the EAT signal peptide by Pvc15 homologs, and subsequent EAT loading into the eCIS particle (Jiang et al, 2022). We combined three features into a single convenient SVM score which separated the EATs from non-EATs (Fig. 4A,B) and is in correlation with the DUF4157 eCIS marker domain (Fig. 4C). Experimental validation of the N-terminal signal peptide as a predictor of eCIS EAT (or other effector) loading is beyond the scope of this work.

Interestingly, a recent preprint by Steiner-Rebrova et al. (Steiner-Rebrova et al, 2023) also included an analysis of the N-terminal signal peptide of the Afp18 EAT combined with empirical Afp particle-loading experiments. The authors concluded that the minimal eCIS N-terminal signal peptide is 20-aa long and has a high percentage of polar amino acids. The three features that were used in our eCIS signal peptide are all located within the 35 N-terminal aa of EATs and they include polarity (high hydro-philicity) as well as two other features. Steiner-Rebrova et al. developed a logistic regression model with a mean cross-validation accuracy of 93.0%. However, they acknowledge in the methods section of their paper that the model suffers from overfitting since it learned to recognize any sequence that looks like Afp18 EAT signal peptide, since the positive set was enriched in sequences similar to this sequence. Altogether, our model of the eCIS N-terminus score is based on a large and diverse sequence dataset, which led to the

identification of three independent discriminatory features (hydrophilicity, aromaticity, and disorder). Needless to say, future experiments should quantify the predictive power of our eCIS signal peptide model and should include more data on bona fide EATs and non-EAT effectors to improve any model.

We hypothesized that EATs would exhibit a shorter protein length compared to the background group as they need to fit a narrow tube. However, contrary to our expectations, the distribution of protein lengths did not reveal a significant difference between the EAT and non-EAT groups. However, our results are supported by the fact that Jiang et al. could package in PVC eCIS particles even effectors of over 1,100 amino acids (CagA from *H. pylori*) (Jiang et al, 2022). In addition, we identified that DUF4157 is an important domain found in EATs and we speculate that its addition following the eCIS signal peptide would improve EAT recognition and eCIS loading. The function of the DUF4157 domain remains elusive in the context of the eCIS function. For one antibacterial EAT, Cgo1, it was reported that bacterial growth inhibition was independent of the metalloprotease motifs found in DUF4157 (Xu et al, 2022). Our results show that a classic EAT gene is often located in the vicinity of DUF4157-containing genes, even if it does not carry this domain by itself (Fig. 2C). It may suggest that DUF4157 functions as an EAT chaperone which can be located in *cis*, within the EAT protein, or in *trans*, by proteins encoded by nearby genes. In this sense, DUF4157 may function similarly to known T6SS adapters and chaperones, which are responsible for effector folding and stability (Unterweger et al, 2017; Manera et al, 2022; Unterweger et al, 2015; Jana and Salomon, 2019; Allsopp and Bernal, 2023). Given the fact that eCIS often carries toxins (EATs) as effectors including antibacterial ones (e.g., EAT14-16), microbes are expected to handle these proteins with care and in a regulated manner to avoid self intoxication.

While our classification tool achieved high predictive accuracy, there is still a possibility of false negatives or false positives. The classifier's performance heavily relies on the quality and representation of the training dataset, with a relatively limited positive set. Furthermore, the heterologous expression experiments confirmed the cytotoxic activity of half of the selected EAT candidates. However, our empirical positive predictive value (50%) is lower than the reported computed value according to XGBoost (94%) within this small sample size. The lack of cytotoxicity of four candidates can result from these genes being bona fide secreted effectors which are non-toxins, as was previously shown for some eCIS (Ericson et al, 2019) and T6SS effectors (Lin et al, 2017; Si et al, 2017; Zhu et al, 2021; Wang et al, 2015). It is also plausible that the toxicity assay's sensitivity is limited in case we tested for cytotoxicity in the wrong target organism, lacking the toxin target. Another possibility is that we expressed the toxin in the wrong compartment. For example, toxins that should target bacterial outer membrane or peptidoglycan would not kill *E. coli* in the cytoplasm. Indeed, we identified that 68 of the predicted EATs have a peptidoglycan-binding domain (Dataset EV4). Further experimental validations using a broader range of organisms and cell types are warranted and can improve EAT detection and elucidate EATs specificity, and additional studies are needed to unravel the specific mechanisms and targets of these toxins.

Exploring the potential of EATs in various applications, such as antimicrobial drug development or biocontrol strategies against pests, is an intriguing avenue for future research due to their

potentially high specificity when combining tail fiber-based adhesion to specific target cells and EATs which might be target-specific. Assessing the efficacy and safety of EAT-based therapeutics in different contexts can open new avenues for combating infections of different pathogens in clinical and agricultural settings, preventing pests, and treating different tumor types.

In conclusion, the successful identification and validation of novel EATs using our machine-learning approach expands our understanding of the diverse activities of eCIS. The resource of predicted EATs and eCIS signal peptides we developed here provides numerous hypotheses for future investigations of >1000 of naturally occuring eCISs. By pursuing these suggested research avenues, we can deepen our understanding of EATs, their mechanisms of action, leading to advancements in both fundamental biology and practical applications.

# Methods

All samples and their corresponding features were obtained from the Integrated Microbial Genomes and Microbiomes database (IMG) (https://img.jgi.doe.gov/) (Chen et al, 2023). The process involved a series of seven steps, as depicted in Fig. 1.

## Data collection, preprocessing, and compilation of EAT and non-EAT sets

For the classification problem, a training set was constructed consisting of a positive set (EATs) and a negative set (non-EATs). The positive set initially contained 22 known and experimentally validated EATs downloaded from IMG. The negative set is comprised of four groups: (1) eCIS core genes, (2) random non-toxin genes on the eCIS operon scaffold, assuming the number of EATs/genome is very low and hence even inclusion of a few randomly selected EATs in this group would have a negligible effect on the final model, (3) random non-toxin genes from eCIS-encoding ("eCIS+") genomes, excluding the eCIS operon scaffold, and (4) effectors associated with type I-IV secretion systems (from https://bastionhub.erc.monash.edu/) (Wang et al, 2020a). We then augmented the positive set with additional, putative EAT genes, including genes adjacent to the eCIS operon with known toxin Pfam domains. To remove sequence redundancy, highly homologous genes (with over 90% identity) were removed to ensure sequence dissimilarity. The resulting positive set consisted of 146 genes, including 22 validated EATs and 124 genes with a high likelihood of being EATs. The negative set comprised 2,949 genes, including 970, 486, 917, and 576 genes from groups 1–4, respectively, mentioned above (Dataset EV1).

## Features extraction

Two groups of features were computed for each candidate gene: genome-based and protein sequence-based.

### Genomic feature group
**Distance from the last core gene**: Based on our previous work (Geller et al, 2021), which identified 1425 eCIS operons across 1249 bacterial and archaeal genomes, it was observed that EAT genes tend to be located downstream of the eCIS operon. To capture this,

the "distance from the last core gene" feature was calculated for each gene, representing the number of genes between the tested gene and the nearest eCIS operon core gene. Genes outside of the eCIS operon scaffold were assigned a value of $-1$, while genes from inside the operon set were assigned a value of 0.

**"DUF4157 Adjacency" and "DUF4157 Presence"**: DUF4157 is a highly conserved eCIS-associated protein domain with an unknown function that serves as a marker of eCIS (Geller et al, 2021) and is sometimes present in T6SS effectors (Wood et al, 2019). The domain carries a HEXXH motif that is characteristic of many families of metallopeptidases (Hooper, 1994). The "DUF4157 adjacency" feature was defined as a binary indicator, with a value of 1 if the DUF4157 domain was present in two protein-coding genes upstream or downstream of the tested gene, and 0 otherwise. Another DUF4157-related feature, "**DUF4157 presence**", was defined as 1 if the tested protein contains this domain, and 0 otherwise.

**eCIS cloud domain presence**: We previously defined "eCIS cloud domains" as eCIS-associated domains found in fewer than four bacterial or archaeal phyla. These domains are enriched in eCIS operons but are relatively diversified and patchy. Namely, different eCIS operons tend to include different cloud domains, unlike core domains which are conserved across most eCIS operons. We previously noted that cloud domains overlap with many toxin domains (Geller et al, 2021). The presence of cloud domains was determined for each protein, assigning a value of 1 if at least one cloud domain was present in the tested protein, and 0 otherwise.

### Biochemical (protein sequence-based) group of features

**Length**: The protein length in amino acids.

**Canonical signal peptide presence**: Presence of a signal peptide that indicates secretion through a SecYEG channel. Signal-P presence information was extracted from IMG based on the Signal-P program (Emanuelsson et al, 2007) for each protein, with a value of 1 indicating the presence of a signal peptide, and 0 indicating its absence. We hypothesized that EATs should not include the canonical bacterial signal peptide but a custom one.

**N-terminal biochemical properties**: Several biochemical properties were calculated for the N-terminal region of each set (EATs or non-EATs). The amino acid range for each feature was determined based on feature calculation for different window sizes (explained below). These specific windows were selected based on a prior analysis that assessed various window lengths, ranging from 0 to 45 amino acids, for each property. These properties included:

**Hydrophobicity of first 35 amino acids:** The Kyte–Doolittle hydrophobicity scale (Kyte and Doolittle, 1982) was employed to compute this feature. Widely utilized for identifying hydrophobic segments within proteins, the Kyte–Doolittle scale assigns a positive value to hydrophobic regions. Within this scale, each amino acid is assigned a numerical value ranging from $-4.5$ to $4.5$ (Dataset EV2), and the cumulative summation of these values across the initial 50 amino acids of each protein was utilized to calculate the index.

**Molecular weight of the first eight amino acids**

**Charge of the 6th–21th N-terminal residues:** The protein charge was computed by aggregating the charges of amino acids situated within residues at positions 6 through 21, at the N-terminal end of the protein. To achieve this, amino acids R, H, and K were assigned a value of 1 to denote positive charge, whereas amino acids D and E were assigned a value of -1 to represent negative charge (Dataset EV2). The remaining amino acids were designated a charge value of 0.

**Number of aromatic amino acids between the first eight N-terminal residues:** a count of phenylalanine (F), tryptophan (W), and tyrosine (Y).

**Disorder-promoting index in the first 6–21 N-terminal residues:** This feature was computed by aggregating the order values of N-terminal amino acids at positions 6 to 21 in each protein of the dataset. The order values correspond to a coding system utilized to represent the propensity of amino acids to facilitate or impede protein folding. Specifically, amino acids A, Q, S, E, R, K, G, and P are acknowledged as promoters of disorder, while I, L, M, V, F, Y, and C are recognized for their role in promoting protein folding (Campen et al, 2008). Consequently, amino acids from the former group were assigned a value of 1, while those from the latter were assigned a value of -1 (Dataset EV2). This encoding strategy enabled us to quantitatively characterize this feature, whereby higher feature values correlated with increased disorder propensity.

Python scripts were developed to compute these characteristics and results are kept in Dataset EV2. The determination of the precise window (defined by the start and end indices within the protein sequence) for each feature was accomplished through the optimization of the Kolmogorov–Smirnov score across a range of subsequences from the N-terminal region of each protein. The optimal window selection process involved the feature value calculation for a minimum window length of five amino acids and a maximum limit of 50 amino acids. For each defined window, the distributions of feature values within the positive and negative sets were compared, and the most statistically significant windows were selected.

### eCIS Signal peptide score calculation

We examined various sequence-based biochemical features to infer their efficacy in distinguishing between secreted EATs and non-EATs. Following this analysis, we identified three N-terminal sequence-based features as particularly relevant: hydrophobicity, sequence disorder, and aromaticity. Subsequently, we trained an SVM classifier with a linear kernel to differentiate between features representing EATs and non-EATs N-terminal peptides (Fig. EV1). Motivated by this separation, we introduced a scoring system for signal peptides (Dataset EV1), based on the posterior probability of each N-terminal peptide to originate from EATs or from non-EATs. This score serves as an indicator of the likelihood of a protein to get loaded into an eCIS particle. A detailed description of the process is provided in the Results section under "Biochemical Features Characterizing EATs and Their Usage in Prediction of eCIS Signal Peptide".

### Model selection

Commonly used machine-learning classification algorithms, including Support Vector Machine (SVM), Logistic Regression, Decision Tree, K-Nearest Neighbors, and Random Forest, were tested. Sensitivity and positive predictive value metrics were used to evaluate the performance

of these algorithms through a fivefold cross-validation training and testing process. An ensemble of decision tree classifiers, trained using the XGBoost boosting algorithm, outperformed the other models significantly, across held-out validation samples. In boosting techniques, all decision trees are constructed from the same data, but the samples are weighted such that each tree assigns higher weights to samples incorrectly classified by the previous trees. Namely, each tree is trained to correct the typical errors made by previous trees. Such ensemble models, trained with boosting techniques, demonstrated stability, robustness, and strong classification capabilities, making them well-suited for our imbalanced dataset (More and Rana, 2017). Hyperparameter tuning was performed to optimize the ensemble performance, ensuring minimal false positive and false negative values while maintaining stability and avoiding overfitting. Optimized hyperparameters contain the number of trees per classifier ($n = 40$), their maximum depth ($n = 7$), and minimum leaf size ($n = 4$). We used the XGBoost Python package, with optimal hyper-parameters (Chen and Guestrin, 2016). The code for the model is publicly available here: https://github.com/AleksaDanube/eCIS_ML

## Model performance evaluation

To evaluate the trained XGBoost model's performance, a fivefold cross-validation approach was employed. The dataset was randomly divided into stratified subsets, with each subset containing 20% of the data. The model was trained five times, using four subsets (a total of 80%) for training and the remaining 20% used for testing. This process allowed the classification of each data sample as either an EAT or a non-EAT. Performance metrics, including positive predictive value and sensitivity, were calculated based on the entire dataset predictions.

$$Sensitivity\ (SS) = \frac{TP}{(TP + FN)}$$

$$Positive\ Predictive\ Value\ (PPV) = \frac{TP}{(TP + FP)}$$

The positive predictive value (PPV, also called precision) metric was deemed particularly significant because we aimed to optimize resource allocation for the subsequent stages of experimental laboratory validation. Namely, we aimed to have a high success rate in the validation experiments. Therefore, maximizing PPV was our primary focus during the training and testing process, rather than focusing on sensitivity.

To ensure the absence of overfitting and the robustness of the model, 19,010 randomly selected genes from eCIS+ genomes, that were not used during the training/testing process were extracted (newly extracted genes). The algorithm was trained 100 times on randomly selected 80% of the dataset (a stratified partition, where 80% were selected from each of the classes separately). Additionally, one training iteration was performed using the entire dataset. This yielded 101 differently trained models, which were used to classify the 19,010 newly extracted genes. The classifications obtained from the 101 models showed a strong consensus, demonstrating the absence of overfitting and the stability of the selected algorithm (Dataset EV3).

## Classification of new EATs

The trained model was applied to all genes obtained from genomes with eCIS presence (eCIS+ genomes). Features of these genes were extracted from IMG, and the trained model was employed to predict their classifications as EAT or non-EAT based on the calculated score with a score >0.5 pointing to an EAT prediction (Dataset EV4).

## Selection of candidates for experimental validation

The selection of candidates for experimental validation was based on factors such as the probability of being an EAT based on our XGboost machine-learning model, novelty (the toxin is unknown), domain architecture, and protein length (prioritizing short proteins). We prioritized promising candidates' EATs based on their high model probability score (>0.93) and filtered based on short length to reduce DNA synthesis costs. We also prioritized proteins of unknown function that contain at least one domain of unknown function (DUF). Eventually, eight EAT candidates were selected for experimental validation in toxicity assays in bacteria and/or yeast (Table 1).

## Experimental validation

### Bacterial strains and strain construction
Candidate EAT gene sequences were retrieved from IMG, synthesized with codon optimized for expression in *E. coli*, and cloned into pBAD24 (a gift from Yechiel Shai, Weizmann Institute of Science) by Twist Bioscience (South San Francisco). The growth control for the experiments was an empty vector. The plasmid was then transformed into *E. coli* BL21 (DE3) strain, using the previously described TSS method (Chung et al, 1989).

### Bacterial heterologous expression
For bacterial drop assays and growth, overnight cultures of the strains harboring the vectors were grown in LB supplemented with 100 mg/liter ampicillin (Tivan biotech) with shaking (225 rpm). The cells were grown on LB plates or in liquid LB with ampicillin (100 mg/ml) and 0.2% arabinose as an inducer or 1% glucose as a repressor for cells containing the toxin candidates. For drop assays, to measure toxicity, cultures were normalized to $OD_{600} = 0.3$ and then serially diluted by a factor of ten. Dilutions were spotted as three biological replicates on LB agar containing inducers or repressors and antibiotics as described above. The plates were incubated overnight at 37 °C. Results were documented using Amersham ImageQuant 800.

### Yeast gene synthesis, heterologous expression, and drop assay
Genes sequences of the putative toxins in yeast were retrieved from IMG. The genes were synthesized, followed by codon adaptation to yeast by Twist Bioscience, and were cloned into pESC-leu galactose inducible plasmids. The plasmids were then transformed into *Saccharomyces Cerevisiae* BY4742 strain. Overnight cultures of the strains containing the vectors of interest were grown in SD-Leu media. For heterologous expression, the culture OD was normalized to $OD_{600} = 0.3$, washed once with water, and then serially diluted by a factor of ten. Dilutions were spotted as three biological

replicates on SD-leu agar containing 2% glucose as a repressor or 2% galactose as an inducer, and plates were incubated for 48 h at 30 °C. Results were documented using Amersham ImageQuant 800.

### Prediction of the active site and construction of mutants
The structural predictions for EAT toxins were conducted through Alphafold2 (Jumper et al, 2021). The predicted structures underwent analysis using foldseek and were cross-referenced with the PDB100 database (van Kempen et al, 2023). To identify potential active sites, we initially searched for matches in the PDB database. In cases where a match was not found, alternative approaches were pursued. Annotations such as Pfam domains were examined, and InterPro was employed to identify signature residues, which were then compared with our protein (Paysan-Lafosse et al, 2023). For a comprehensive presentation of the matched structures between our EATs and proteins from the PDB, ChimeraX facilitated 3D visualization (Meng et al, 2023).

To create gene mutants, we used the New England Biolabs (NEB) advanced site-directed mutagenesis technique, utilizing the NEBuilder® HiFi DNA Assembly Master Mix. Briefly, primers were designed to incorporate the desired amino acid mutations, yielding two PCR fragments that include the point mutation within the overlapping region. Subsequently these were amalgamated during the construct assembly reaction. To verify its accuracy, the resulting construct was sequenced. Following sequence confirmation, the freshly engineered plasmids were introduced into BL21 cells, initiating the subsequent phase of heterologous expression experiments. The primers are listed in Appendix File 1.

## Expression of EATs in HEK293T cells

### Plasmid construction
Mammalian expression vectors encoding the genes for VopC or EATs (14-17) were constructed utilizing eBlock gDNA fragments based on accession numbers: VopC, WP_005463254.1; EAT14, WP_162277501.1; EAT15, WP_034246516.1; EAT16, WP_082303719.1; and EAT17, WP_013435507.1. The fragments were joined together and ligated into the pcDNA3.1 plasmid. The resulting plasmid constructs were sequence-confirmed and transformed into *E. coli* TOP10 cells. Plasmid DNA used for transfection was purified from corresponding bacterial cultures using GeneJET Plasmid Miniprep Kit (Thermo Fisher Scientific).

### Activity assays in mammalian cells
HEK293T cells (ATCC number CRL-11268) were cultured in Dulbecco's modified Eagle's medium (DMEM; Gibco, Invitrogen) supplemented with 0.37% sodium bicarbonate, 100 units/mL penicillin-streptomycin (Thermo Fisher Scientific), and 5% HyClone bovine growth serum (BGS) in 100-mm culture dishes. We did not recently check the HEK293T cells for Mycoplasma contamination, but it is tested according to the product manual. The cell lines were not recently authenticated. At 50% confluency, cells were replated into six-well plates after trypsinization using TrypLE™ express enzyme (Thermo Fisher Scientific) and incubated for ~12 h before transfection. At 80% confluency, cells were transfected with two to four plasmids as indicated using the calcium phosphate method, as previously described (Haywood et al, 2018, 2021). Each transfection mixture contained an empty-vector control or the indicated VopC- or EAT-encoding plasmids, alone

or in combination at the indicated ratio, in addition to the two reporter gene plasmids: a SRE promoter fused to a firefly luciferase reporter gene (pSRE-luc, Stratagene), and a herpes simplex virus TK promoter fused to a Renilla luciferase gene, as a low-expression constitutive reporter control gene (pGL4.74 hRluc/TK, Promega Madison, WI) at a final DNA concentration in each well of 2.5 µg/ml pSRE-luc and 0.25 µg/ml pGL4.74 hRluc/TK, respectively. Cells were incubated for 7 h, and then each transfection was replated into four wells of 48-well plates with fresh 2% BGS DMEM and incubated for 17 h at 37 °C, after which the medium was removed, and the cells were lysed with 50 µl of Passive Lysis Buffer (Promega). After 10 min incubation on a rocker, lysates from each well was transferred to a 96-well plate and analyzed for firefly luciferase reporter activity and Renilla luciferase control activity using the Promega Dual-Luciferase® Reporter 1000 Assay System by addition of 25 µl of Luciferase Assay Reagent followed by 25 µl of Stop and Glo Buffer per well, according to the manufacturer's protocol. Luminescence was measured using a Synergy-HT multi-detection microplate reader (BioTek Winooski, VT), and results were generated using the BioTek microplate software Gen5 and reported as relative light units (RLUs) with the following settings: sensitivity, 108; integration time, 0.7 s. Each experiment was repeated at least 3 times in quadruplicate. Luciferase activity was determined by dividing the firefly RLUs by the Renilla RLUs. The relative luciferase activity was determined by dividing the luciferase activity for experimental samples by the mean luciferase activity for the empty vector control (EV) samples of each repeat. All necessary processing, calculation of initial data, and visualizations were performed through R language and R libraries (ggplot2, reshape, ggpubr).

Blinding was not done as part of the study (irrelevant).

## Data availability

Full training set and EAT predictions are available as expanded view (EV) Datasets. The machine-learning model produced in this study is based on the public XGBoot python package and is available as follows:

- Machine-learning model: GitHub (https://github.com/AleksaDanube/eCIS_ML)

The source data of this paper are collected in the following database record: biostudies:S-SCDT-10_1038-S44320-024-00053-6.

## Peer review information

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

## Acknowledgements

A.L. is supported by the Israeli Science Foundation (Grant Nos. 1535/20, 3300/20, and 3062/20), the Volkswagen Foundation (Grant ZN 4041), and the Hebrew University of Jerusalem and University of Illinois System Joint Research and Innovation Seed Grant (to B.A.W. and A.L.). B.A.W. is also supported by a Grant RB#18122 from the Research Board of the University of Illinois at Urbana-Champaign. We thank Alexander Geller for careful reading of the manuscript. The synopsis figure was done using Biorender.

## Author contributions

**Aleks Danov**: Data curation; Software and Algorithm Development; Investigation; Visualization; Methodology; Writing—original draft. **Inbal Pollin**: Validation; Investigation; Visualization; Methodology; Writing—original draft. **Eric Moon**: Methodology. **Mengfei Ho**: Supervision; Visualization; Methodology. **Brenda, A Wilson**: Supervision; Funding acquisition; Investigation; Visualization; Writing—original draft. **Philippos A Papathanos**: Supervision; Funding acquisition; Writing—original draft. **Tommy Kaplan**: Conceptualization; Software Development; Supervision; Funding acquisition; Visualization; Methodology. **Asaf Levy**: Conceptualization; Data curation; Supervision; Funding acquisition; Investigation; Visualization; Writing—original draft; Project administration; Writing—review and editing.

Source data underlying figure panels in this paper may have individual authorship assigned. Where available, figure panel/source data authorship is listed in the following database record: biostudies:S-SCDT-10_1038-S44320-024-00053-6.

## Disclosure and competing interests statement

The authors declare no competing interests.

# Expanded View Figures

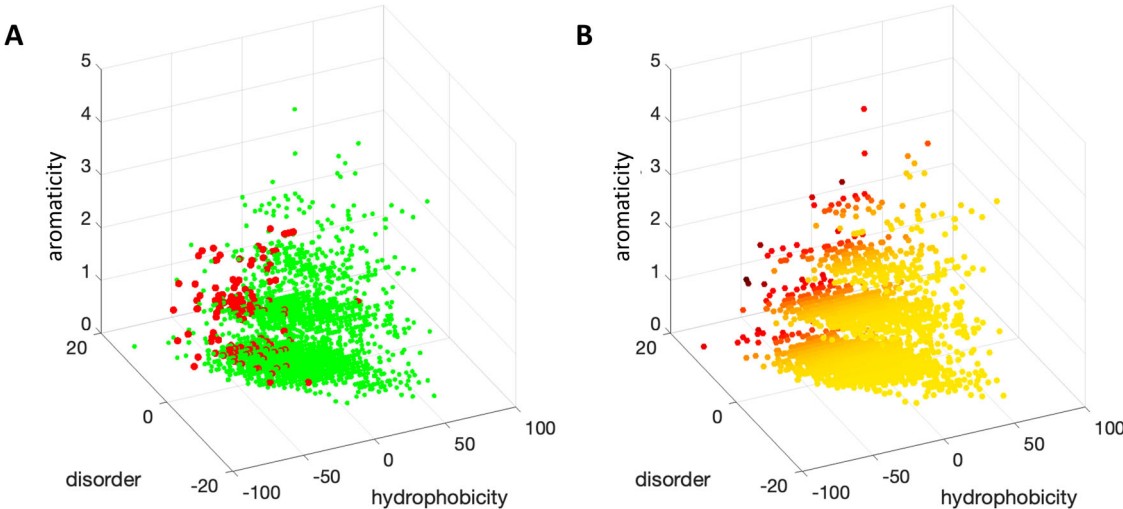

**Figure EV1.  The scatterplot of three sequence-based features that were used to encode eCIS signal peptide score based on linear SVM.**

(**A**) The red dots are EATs and the green dots are non-EATs. (**B**) The colormap is defined by the score values with red being high values and yellow being near zero values. The plots show the high consistency between the positive set and the higher score values.

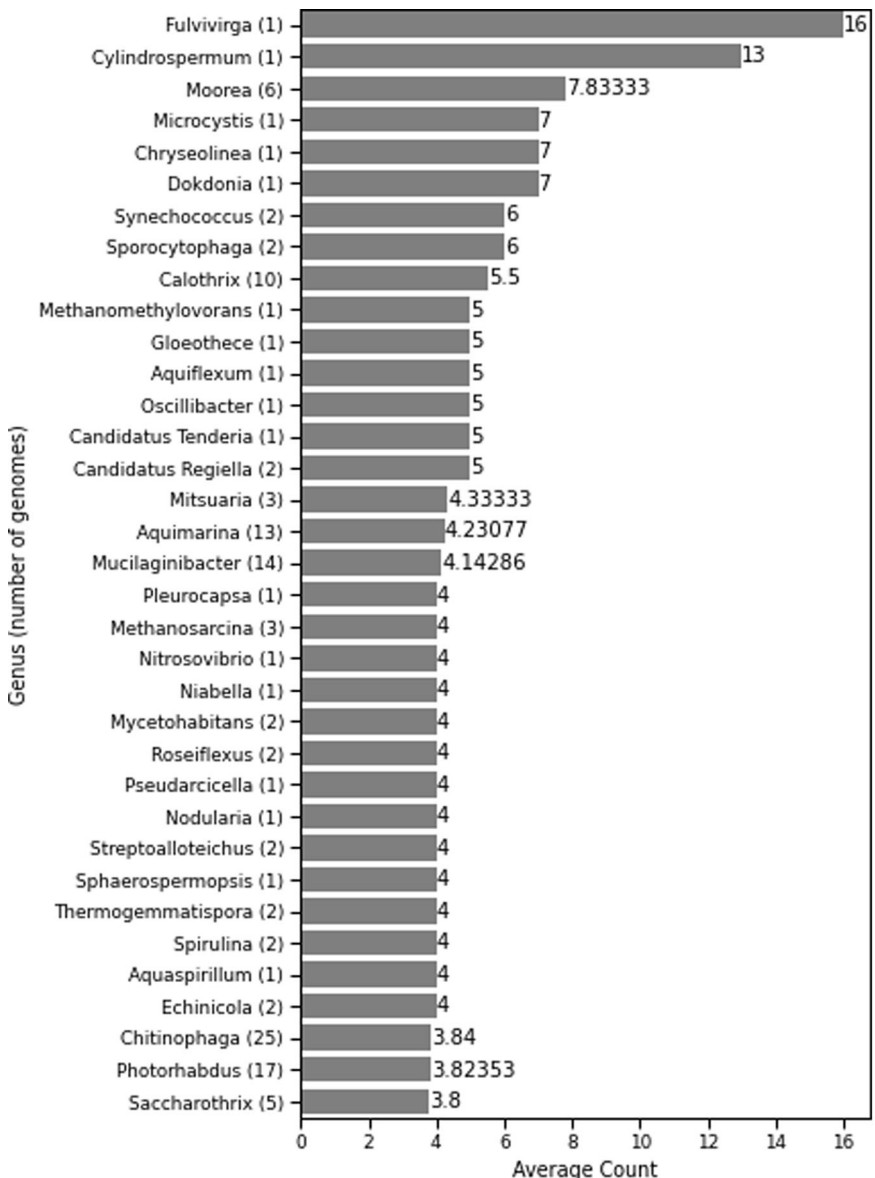

**Figure EV2.  Average number of predicted EATs per genus.**

In parenthesis is the number of genomes per genus in our dataset.

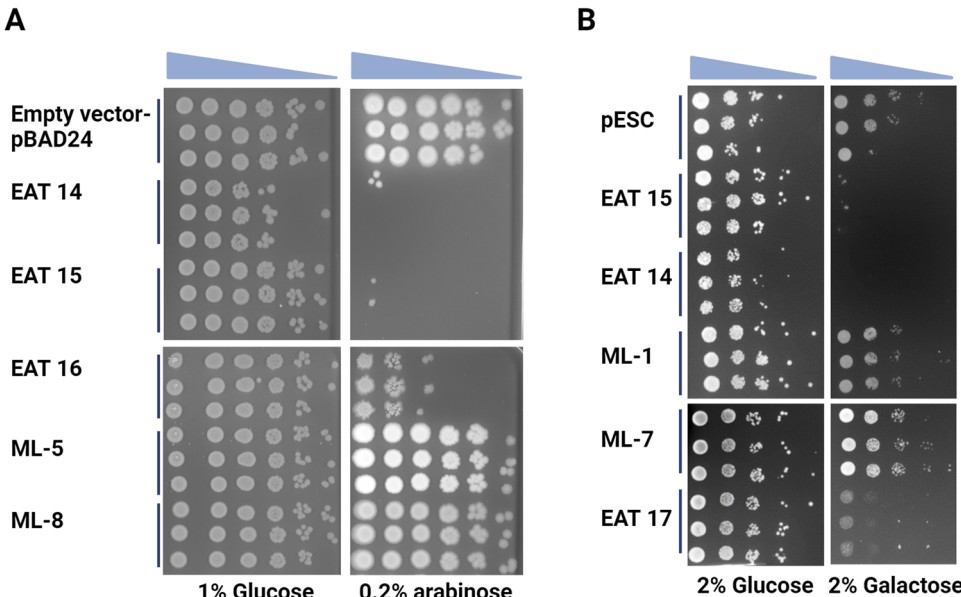

**Figure EV3.   Drop assays including the predicted effectors that were not toxic.**

(A) Bacterial drop assay. Induced with 0.2% Arabinose and uninduced in the presence of 1% Glucose. Light blue triangles indicate the serial dilution of the droplets.
(B) Yeast drop assay, induced with 2% Galactose and inhibited with 2% Glucose. For the negative control in the yeast drop assay, a pESC empty vector was used. Candidates who did not work maintained the annotation ML-X (ML- preadicted by Machine Learning).

