## [Peer Review File · Molecular Systems Biology]

Genome-wide discovery of toxins associated with the extracellular contractile injection system

Aleks Danov, Inbal Pollin, Eric Moon, Mengfei Ho, Brenda Wilson, Philipos Papathanos, Tommy Kaplan, and Asaf Levy

Corresponding author(s): Asaf Levy (alevy@mail.huji.ac.il)

Review Timeline:

Submission Date:	24th Jan 24
Editorial Decision:	8th Feb 24
Appeal Received:	13th Feb 24
Editorial Decision:	9th Apr 24
Revision Received:	25th Apr 24
Editorial Decision:	23rd May 24
Revision Received:	6th Jun 24
Accepted:	27th Jun 24

Editor: Poonam Bheda

Transaction Report:

8th Feb 2024

RE: Manuscript MSB-2024-12239, Genome-wide discovery of toxins associated with the extracellular contractile injection system

Dear Dr Levy,

Thank you for having submitted a manuscript entitled "Genome-wide discovery of toxins associated with the extracellular contractile injection system" for consideration for publication in Molecular Systems Biology. I have now had the chance to read your study and I have discussed it with the team. We have also consulted an expert in the field and I am afraid that the outcome was not positive.

In this study, you developed a machine learning classifier to identify extracellular contractile injection system (eCIS)-associated toxins (EATs) and using this classifier you identified ~2,000 genes from 950 genomes as putative EATs. You then experimentally tested and validated 4 out of 8 EATs and identified key enzymatic residues through structure prediction and mutation. While I appreciate the more systematic approach to identify new EATs with your classifier, editorially we agreed that the findings are somewhat limited, also in light of your recent Nature Communications paper. Please note that we have also consulted an expert in the field, who thought that the overall biological insight remains somewhat limited.

Taken together and based on the recommendation we received from an expert in the field we unfortunately have decided to not send the study out for review. That said, your work is a good candidate for Life Science Alliance (<http://www.life-science-alliance.org/>) our broad scope Open Access journal published in partnership between EMBO Press, Rockefeller University Press, and Cold Spring Harbor Laboratory Press. The editors of Life Science Alliance would be pleased to send your manuscript for peer review. Please use the following link to transfer your manuscript (no reformatting is required): Link Not Available

Eric Sawey, executive editor of Life Science Alliance (e.sawey@life-science-alliance.org), would be happy to answer any questions.

I apologize for not bringing better news regarding the publication of your study at Molecular Systems Biology and I hope that you will view the possibility of a transfer to Life Science Alliance favorably.

Kind regards,

Poonam

Poonam Bheda, PhD
Scientific Editor
Molecular Systems Biology

=====

** As a service to authors, EMBO Press offers the possibility to directly transfer declined manuscripts to another EMBO Press title or to the open access journal Life Science Alliance launched in partnership between EMBO Press, Rockefeller University Press and Cold Spring Harbor Laboratory Press. The full manuscript and if applicable, reviewers' reports, are automatically sent to the receiving journal to allow for fast handling and a prompt decision on your manuscript. For more details of this service, and to transfer your manuscript please click on Link Not Available. **

Dear Dr. Bheda,

Thank you for your decision which is a bit frustrating. It feels like a misunderstanding. You mentioned: "editorially we agreed that the findings are somewhat limited, also in light of your recent Nature Communications paper."

The focus of the current paper is very different from the previous Nature Comm. paper which characterized mostly eCIS operons and their genomic distribution. We predicted the toxins in that paper manually.

In comparison, in the current work:

1. This is the first work to provide the community EAT gene predictions, therefore I believe it will be highly cited (our previous paper is already cited 33 times). Our validation experiments showed some of the results with some extensive analysis of the biology of 4 EATs.
2. This work provides the first classifier of the N terminal signal peptide of eCIS, which can be added to any protein to allow efficient loading into different types of eCIS.
3. Combined, we show the important features in both an EAT gene and its signal peptide.

In my view this is a key resource to the growing community studying the eCIS secretion system, especially since the recent Krietz et al. Nature 2023 paper which showed the biotechnological applications of eCIS.

I would appreciate it if you'd reconsider your decision.

Best regard,
Asaf

9th Apr 2024

Manuscript Number: MSB-2024-12239R-Q

Title: Genome-wide discovery of toxins associated with the extracellular contractile injection system

Dear Dr Levy,

Thank you again for submitting your work to Molecular Systems Biology, and I once again sincerely apologize for the extraordinary delay in sending you a decision on your work. We have now heard back from the three reviewers who agreed to evaluate your study. As you will see below, the reviewers appreciate that the presented approach and experimental validation of EATs. However, Reviewer 1 raises a few points on the high prediction accuracy given the limited statistical power and the bioinformatics analyses of the experimentally validated EATs that we would ask you to address in a minor revision.

We require:

1) A .docx formatted version of the manuscript text (including legends for main figures, EV figures and tables). Please make sure that the changes are highlighted to be clearly visible. Alternatively you may choose to submit your manuscript as a LaTeX file.

4) A .docx formatted letter INCLUDING the reviewers' reports and your detailed point-by-point responses to their comments. As part of the EMBO Press transparent editorial process, the point-by-point response is part of the Peer Review File (PRF), which will be published alongside your paper.

5) A complete author checklist, which you can download from our author guidelines (<https://www.embopress.org/page/journal/17574684/authorguide#submissionofrevisions>). Please insert information in the checklist that is also reflected in the manuscript. The completed author checklist will also be part of the PRF.

6) Please note that all corresponding authors are required to supply an ORCID ID for their name upon submission of a revised manuscript.

7) It is mandatory to include a 'Data Availability' section after the Materials and Methods. Before submitting your revision, primary datasets produced in this study need to be deposited in an appropriate public database, and the accession numbers and database listed under 'Data Availability'. Please remember to provide a reviewer password if the datasets are not yet public (see <https://www.embopress.org/page/journal/17574684/authorguide#dataavailability>).

In case you have no data that requires deposition in a public database, please state so in this section. Note that the Data Availability Section is restricted to new primary data that are part of this study. This study includes no data deposited in external repositories.

8) For data quantification: please specify the name of the statistical test used to generate error bars and P values, the number (n) of independent experiments (specify technical or biological replicates) underlying each data point and the test used to calculate p-values in each figure legend. The figure legends should contain a basic description of n, P and the test applied. Graphs must include a description of the bars and the error bars (s.d., s.e.m.). Please provide exact p values.

<https://www.embopress.org/page/journal/17574684/authorguide#expandedview>

11) For more information: There is space at the end of each article to list relevant web links for further consultation by our readers. Could you identify some relevant ones and provide such information as well? Some examples are patient associations, relevant databases, OMIM/proteins/genes links, author's websites, etc...

12) Author contributions: CRediT has replaced the traditional author contributions section because it offers a systematic machine readable author contributions format that allows for more effective research assessment. Please remove the Authors Contributions from the manuscript and use the free text boxes beneath each contributing author's name in our system to add specific details on the author's contribution. More information is available in our guide to authors.

13) Disclosure statement and competing interests: We updated our journal's competing interests policy in January 2022 and request authors to consider both actual and perceived competing interests. Please review the policy

<https://www.embopress.org/competing-interests> and update your competing interests if necessary.

14) Every published paper now includes a 'Synopsis' to further enhance discoverability. Synopses are displayed on the journal webpage and are freely accessible to all readers. They include a short stand first (maximum of 300 characters, including space) as well as 2-5 one-sentences bullet points that summarizes the paper. Please write the bullet points to summarize the key NEW findings. They should be designed to be complementary to the abstract - i.e. not repeat the same text. We encourage inclusion of key acronyms and quantitative information (maximum of 30 words / bullet point). Please use the passive voice. Please attach these in a separate file or send them by email, we will incorporate them accordingly.

Share synopsis text and image, as well as eTOC:

Please note that these would be the final versions and changes during proofing are usually not allowed

15) As part of the EMBO Publications transparent editorial process initiative (see our policy here:

https://www.embopress.org/transparent-process#Review_Process), Molecular Systems Biology will publish online a Peer Review File (PRF) to accompany accepted manuscripts.

In the event of acceptance, this file will be published in conjunction with your paper and will include the anonymous referee reports, your point-by-point response and all pertinent correspondence relating to the manuscript. Let us know whether you agree with the publication of the PRF and as here, if you want to remove or not any figures from it prior to publication.

Please note that the Authors checklist will be published at the end of the PRF.

Molecular Systems Biology has a "scooping protection" policy, whereby similar findings that are published by others during review or revision are not a criterion for rejection. Should you decide to submit a revised version, I do ask that you get in touch after three months if you have not completed it, to update us on the status.

I look forward to receiving your revised manuscript.

Yours sincerely,

Poonam Bheda

Poonam Bheda, PhD
Scientific Editor
Molecular Systems Biology

Reviewer #1:

The manuscript by Danov et al reports a machine learning approach for prediction of toxins excreted by extracellular contractile injection system (eCIS), known as EATs (eCIS-Associated Toxins).

They develop an EAT classifier and apply it to predict >2000 EATs encoded in nearly a 1000 bacterial and archaeal genomes. The prediction is then followed by experimental validation of 4 distinct EATs.

To the best of my understanding, the feature selection and training of the classifier, which comprise the core of this work, were done appropriately. However, I have a major concern regarding the size of the positive training set of EATs. There were only 22 experimentally validated EATs augmented with 124 additional one predicted with more or less ad hoc approaches. With such a small and not fully validated training set, I am just not convinced by the claims of high prediction accuracy. We do not really know the meaning of the scores reported for the predicted EATs. The experiments reported at the end of the Results appear convincing, but with only 4 EATs tested, have no statistical power.

regarding the specific EATs that have been studied experimentally, I believe that the bioinformatic analysis of these proteins should be somewhat more careful and rely entirely on available annotations. For instance, for EAT14, the authors should take into account the possibility that a transglutaminase family protein could also be a protease which is indeed the likely activity of the toxin. The description of EAT16 is confusing "Its N-terminus contains a DUF4365 and another annotation of Holliday Junction resolvase (HJR)". I assume these are two distinct domains. This EAT seems to be confidently predicted to possess endonuclease activity, the authors should be clear about this.

Reviewer #2:

This manuscript describes an algorithm to predict new effector proteins for eCIS systems. The authors do a good job of describing their thought process and in testing new effectors for antibacterial and/or antifungal activity.

I think the manuscript is quite ready for publication already, there are just a couple of things that I think can be clarified further:

L138: How did you ensure that genes in class 2 were not EATs for the training set? Would just be good to provide a bit more description as to how genes were classified into this group (random non-toxin genes on the eCIS operon scaffold

L230: although you describe analyses of the molecular patterns associated with EATs later in the manuscript, it would be good here to reference this data later (otherwise, the first read through I was left wondering what thresholds and such you picked for the molecular patterns).

Reviewer #3:

This is an interesting report that identifies toxins associated with type VI secretions systems in a wide swath of bacterial species. The authors most important figure on technique is Fig. 5B which summarizes the strategies used to identify and exclude orfs from consideration. The result of this analysis was to whittle candidates down to ones that have domains associated with transglutamatase activity, indicating many may perform posttranslation modification on target proteins.

I think the manuscript is wonderful and have no negative criticisms.

We thank the editor and the three reviewers for taking the time reading the manuscript and their efforts to improve the quality of our manuscript. The revisions we made following the constructive review are highlighted in yellow in the revised file.

Reviewer #1:

The manuscript by Danov et al reports a machine learning approach for prediction of toxins excreted by extracellular contractile injection system (eCIS), known as EATs (eCIS-Associated Toxins).

They develop an EAT classifier and apply it to predict >2000 EATs encoded in nearly a 1000 bacterial and archaeal genomes. The prediction is then followed by experimental validation of 4 distinct EATs.

To the best of my understanding, the feature selection and training of the classifier, which comprise the core of this work, were done appropriately. However, I have a major concern regarding the size of the positive training set of EATs. There were only 22 experimentally validated EATs augmented with 124 additional one predicted with more or less ad hoc approaches. With such a small and not fully validated training set, I am just not convinced by the claims of high prediction accuracy. We do not really know the meaning of the scores reported for the predicted EATs. The experiments reported at the end of the Results appear convincing, but with only 4 EATs tested, have no statistical power.

We were cautious with our claims and specifically mentioned the small validated toxin set (the positive set). Lines 721 in the discussion section read:

“While our classification tool achieved high predictive accuracy, there is still a possibility of false negatives or false positives. The classifier's performance heavily relies on the quality and representation of the training dataset, with a relatively limited positive set.”

We also added now:

“Furthermore, the heterologous expression experiments confirmed the cytotoxic activity of half of the selected EAT candidates. However, our empirical positive predictive value (50%) is lower than the computed value according to XGBoost (94%) within this small sample size.”

regarding the specific EATs that have been studied experimentally, I believe that the bioinformatic analysis of these proteins should be somewhat more careful and rely entirely on available annotations. For instance, for EAT14, the authors should take into account the possibility that a transglutaminase family protein could also be a protease which is indeed the likely activity of the toxin.

We read the literature carefully and this is an interesting idea. It seems that the proteases previously annotated as transglutaminases mostly carry domain PF01841, whereas EAT14 carries domain PF09017, which is more likely acting as a *bona fide* transglutaminase. However, to be on the safe side we added the following sentence:

“Notably, several TG-like proteins have been identified in the past as proteases (Sanchez-Pulido & Ponting, 2016; Ozhelvaci & Steczkiewicz, 2023; Makarova *et al*, 1999), hence we

cannot rule out at the moment the possibility that EAT14 actually serves as a protease and not as a TG. “

The description of EAT16 is confusing "Its N-terminus contains a DUF4365 and another annotation of Holliday Junction resolvase (HJR)". I assume these are two distinct domains. This EAT seems to be confidently predicted to possess endonuclease activity, the authors should be clear about this.

We rephrased the text for EAT16 based on your comment and a new reference:

“EAT16 comprises two distinct domains based on its folding using Alphafold2. Its N-terminus contains a DUF4365 and bears similarity to Holliday Junction resolvase (HJR) (Figure 6H). HJR is a critical enzyme involved in DNA recombination (Aravind *et al*, 2000), suggesting that EAT16 may function as a nuclease or as a DNA binding protein, primarily targeting DNA. DUF4365 was recently shown to act as a nuclease leading to a nonspecific DNA degradation (Lu *et al*, 2024), suggesting a similar mode of action to EAT16.”

Reviewer #2:

This manuscript describes an algorithm to predict new effector proteins for eCIS systems. The authors do a good job of describing their thought process and in testing new effectors for antibacterial and/or antifungal activity.

I think the manuscript is quite ready for publication already, there are just a couple of things that I think can be clarified further:

Thank you very much!

L138: How did you ensure that genes in class 2 were not EATs for the training set? Would just be good to provide a bit more description as to how genes were classified into this group (random non-toxin genes on the eCIS operon scaffold

This is a good point and we may have included a few EATs in this group. We added a comment on L130:

(2) random non-toxin genes on the eCIS operon scaffold, assuming number of EATs/genome is very low and hence even inclusion of a few randomly selected EATs in this group would have a negligible effect on the final model

L230: although you describe analyses of the molecular patterns associated with EATs later in the manuscript, it would be good here to reference this data later (otherwise, the first read through I was left wondering what thresholds and such you picked for the molecular patterns).

Good point. At the end of the paragraph we added the text on L232: “A detailed description of the process is provided in the Results section under “Biochemical Features Characterizing EATs and Their Usage in Prediction of eCIS Signal Peptide”.

Reviewer #3:

This is an interesting report that identifies toxins associated with type VI secretions systems in a wide swath of bacterial species. The authors most important figure on technique is Fig. 5B which summarizes the strategies used to identify and exclude orfs from consideration. The result of this analysis was to whittle candidates down to ones that have domains associated with transglutamatase activity, indicating many may perform posttranslation modification on target proteins.

I think the manuscript is wonderful and have no negative criticisms.

Thank you very much!

23rd May 2024

Manuscript Number: MSB-2024-12239RR

Title: Genome-wide discovery of toxins associated with the extracellular contractile injection system

Dear Dr Levy,

Thank you for the submission of your revised manuscript to Molecular Systems Biology. We have now received the enclosed reports from the referees that were asked to re-assess it. As you will see the reviewers are now globally supportive and I am pleased to inform you that we will be able to accept your manuscript pending the following final amendments:

- 1) Please check the "Author Checklist" carefully and complete all relevant questions. Currently information in the "Reporting" section is missing. If these are not applicable to your study, please select this as a response.
- 2) We require an institutional email address in the manuscript and in our submission system for corresponding authors - currently the manuscript does not have an institutional email address listed for Dr. Levy.
- 3) In the main manuscript file, please include keywords to max. 5.
- 4) Please upload your machine learning model to a suitable repository such as Biomodels or Github to ensure that it is freely publicly available. Please also include a README file with practical use instructions for potential future users of your code/model.
- 5) Please include a Data availability section describing how the data, code etc. have been made available. This section needs to be formatted according to the example below:

"The datasets and computer code produced in this study are available in the following databases:

- Modeling computer scripts: GitHub (<https://github.com/SysBioChalmers/GECKO/releases/tag/v1.0>)

- [data type]: [full name of the resource] [accession number/identifier] ([doi or URL or identifiers.org/DATABASE:ACCESSION])"

- 6) Please include a "Disclosure and competing interests statement". We updated our journal's competing interests policy in January 2022 and request authors to consider both actual and perceived competing interests. Please review the policy <https://www.embopress.org/competing-interests> and update your competing interests if necessary.

- 7) In the Materials and Methods, please take care of the following:

- Cell lines: Please be sure to include a sentence in the Materials and Methods as to whether or not the cell lines were recently authenticated and tested for mycoplasma contamination. Please also update the Author Checklist with the information as to which section in the manuscript this statement has been included.
- Primers: please ensure primers used are included in the Materials and Methods (or if included in table format, that the table is included in an Appendix file). Currently you have indicated that primers were designed to create mutations, but the sequences were not provided. Please also be sure to update the Author Checklist with where this information is provided. If you provide these in an Appendix file, please upload the Appendix as a single PDF (no separate image files are needed) and include page numbers in a Table of Contents. Please also ensure the word "Appendix" is included in all labels for Appendix Figures and Appendix Tables including in the Table of Contents.
- Please ensure that a statement on whether or not blinding was done is included in the Materials and Methods even if no blinding was done. Please also update the Author Checklist with this information.

- 8) Please place individual sections of the manuscript in the following order: Title page - Abstract & Keywords - Introduction - Results - Discussion - Materials & Methods - Data Availability - Acknowledgements - Disclosure and Competing Interests Statement - References - Figure Legends - Expanded View Figure Legends.

- 9) For the figures and figure legends, please take care of the following:

- Figures 2, 3 and 4 are uploaded as SVG files, which we do not accept. Please upload these files as .eps, .tif, or .jpg formats.
 - Please make sure to update the callouts of all figures in the main manuscript text (currently figure callouts are missing for Figure 2D, Figure 7F; callouts for Figure 8A and 8B present but the labels are missing in the figure)
 - The yeast colony images in Figure 1 are re-used in Figure 6B. This is fine as Figure 1 is a summary figure, but please mention this in the respective figure legends.
 - Please note that the figure panels of figures 8a-b are not labelled in the figure. This needs to be rectified.
- Please note that the legends for figures 7c-f is not provided in the sequential manner (legend for figure 7d, g is provided before legend of figure 7b-c, 7e-f; and legend for figure 7e, h is provided before legend of figure 7c, f). This needs to be rectified.
- Please define the annotated p values ***/** in the legend of figure 8a-b as appropriate.
 - Please indicate the statistical test used for data analysis in the legends of figures 8a-b.
 - Please note that information related to n is missing in the legends of figures 8a-b.
 - Please note that the error bars are not defined in the legends of figures 8a-b.

- 10) Tables: The 4 EV tables should be renamed and uploaded as Datasets EV1-EV4; the callouts in the manuscript also need to be corrected accordingly as well as the source file names (currently they are labeled as Supplementary Tables 1-4). Each dataset will need its legend removed from the manuscript and added to the corresponding file in a separate tab.

- 11) Funding: Please ensure that all funding sources are entered into the manuscript submission system (i.e. please add - 3062/20 grant number for Israeli Science Foundation and the Innovation Seed Grant)

12) Synopsis:

- Synopsis image: Please include a graphic that summarises the main findings of the manuscript on a glance and upload it as a high-resolution jpeg file 550 pixels wide x (250-400) pixels high.
- Synopsis text: Please provide a short standfirst (maximum of 300 characters, including space), limit the bullet points to max. 5 and upload it as a separate .doc file. Please write the bullet points to summarise the key NEW findings. They should be designed to be complementary to the abstract - i.e. not repeat the same text. We encourage inclusion of key acronyms and quantitative information (maximum of 30 words / bullet point). Please use the passive voice and upload this as a separate file.
- Please check your synopsis text and image before submission with your revised manuscript. Please be aware that in the proof stage minor corrections only are allowed (e.g., typos).

13) Source Data: Please ensure that a completed Source Data checklist is uploaded (sent to you by my colleague Hannah Sonntag). Please also ensure that the source data are uploaded as a single source data file (zipped) per figure, with the panels clearly visible in the folder structure.

14) As part of the EMBO Publications transparent editorial process initiative (see our policy here: https://www.embopress.org/transparent-process#Review_Process), Molecular Systems Biology will publish online a Peer Review File (PRF) to accompany accepted manuscripts. This file will be published in conjunction with your paper and will include the anonymous referee reports, your point-by-point response and all pertinent correspondence relating to the manuscript. Let us know whether you agree with the publication of the PRF and as here, if you want to remove or not any figures from it prior to publication. Please note that the Authors checklist will be published at the end of the PRF.

15) Please provide a point-by-point letter INCLUDING my comments as well as the reviewer's reports and your detailed responses (as Word file).

I look forward to reading a new revised version of your manuscript as soon as possible.

Yours sincerely,

Poonam Bheda, PhD
Scientific Editor
Molecular Systems Biology

Reviewer #1:

The authors have adequately addressed all the comments of the reviewers. It is truly valuable work. No further comments.

MSB-2024-12239RR - revisions

Title: Genome-wide discovery of toxins associated with the extracellular contractile injection system

Asaf Levy's response is in blue.

Reviewer #1:

The authors have adequately addressed all the comments of the reviewers. It is truly valuable work. No further comments.

Thank you.

1) Please check the "Author Checklist" carefully and complete all relevant questions. Currently information in the "Reporting" section is missing. If these are not applicable to your study, please select this as a response.

Done.

2) We require an institutional email address in the manuscript and in our submission system for corresponding authors - currently the manuscript does not have an institutional email address listed for Dr. Levy.

Done.

3) In the main manuscript file, please include keywords to max. 5.

Done.

4) Please upload your machine learning model to a suitable repository such as Biomodels or Github to ensure that it is freely publicly available. Please also include a README file with practical use instructions for potential future users of your code/model.

Done. The code for the model is publicly available here:

https://github.com/AleksaDanube/eCIS_ML

5) Please include a Data availability section describing how the data, code etc. have been made available. This section needs to be formatted according to the example below:

"The datasets and computer code produced in this study are available in the following databases:

- Modeling computer scripts: GitHub

(<https://github.com/SysBioChalmers/GECKO/releases/tag/v1.0>)

- [data type]: [full name of the resource] [accession number/identifier] ([doi or URL or identifiers.org/DATABASE:ACCESSION)]"

Done. The link to GitHub appears in the M&M and Data Availability sections.

6) Please include a "Disclosure and competing interests statement". We updated our journal's competing interests policy in January 2022 and request authors to consider both actual and perceived competing interests. Please review the policy <https://www.embopress.org/competing-interests> and update your competing interests if necessary.

Done.

7) In the Materials and Methods, please take care of the following:

- Cell lines: Please be sure to include a sentence in the Materials and Methods as to whether or not the cell lines were recently authenticated and tested for mycoplasma contamination. Please also update the Author Checklist with the information as to which section in the manuscript this statement has been included.

Done.

- Primers: please ensure primers used are included in the Materials and Methods (or if included in table format, that the table is included in an Appendix file). Currently you have indicated that primers were designed to create mutations, but the sequences were not provided. Please also be sure to update the Author Checklist with where this information is provided. If you provide these in an Appendix file, please upload the Appendix as a single PDF (no separate image files are needed) and include page numbers in a Table of Contents. Please also ensure the word "Appendix" is included in all labels for Appendix Figures and Appendix Tables including in the Table of Contents.

Done

- Please ensure that a statement on whether or not blinding was done is included in the Materials and Methods even if no blinding was done. Please also update the Author Checklist with this information.

This is 100% irrelevant to the paper.

8) Please place individual sections of the manuscript in the following order: Title page - Abstract & Keywords - Introduction - Results - Discussion - Materials & Methods - Data Availability - Acknowledgements - Disclosure and Competing Interests Statement - References - Figure Legends - Expanded View Figure Legends.

Done.

9) For the figures and figure legends, please take care of the following:

- Figures 2, 3 and 4 are uploaded as SVG files, which we do not accept. Please upload these files as .eps, .tif, or .jpg formats.

Done.

- Please make sure to update the callouts of all figures in the main manuscript text (currently figure callouts are missing for Figure 2D, Figure 7F; callouts for Figure 8A and 8B present but the labels are missing in the figure)

Done.

- The yeast colony images in Figure 1 are re-used in Figure 6B. This is fine as Figure 1 is a summary figure, but please mention this in the respective figure legends.

Done.

- Please note that the figure panels of figures 8a-b are not labelled in the figure. This needs to be rectified.

Done.

Please note that the legends for figures 7c-f is not provided in the sequential manner (legend for figure 7d, g is provided before legend of figure 7b-c, 7e-f; and legend for figure 7e, h is provided before legend of figure 7c, f). This needs to be rectified.

Done.

- Please define the annotated p values $^{***}/^{**}$ in the legend of figure 8a-b as appropriate.
- Please indicate the statistical test used for data analysis in the legends of figures 8a-b.
- Please note that information related to n is missing in the legends of figures 8a-b.
- Please note that the error bars are not defined in the legends of figures 8a-b.

Done.

10) Tables: The 4 EV tables should be renamed and uploaded as Datasets EV1-EV4; the callouts in the manuscript also need to be corrected accordingly as well as the source file names (currently they are labeled as Supplementary Tables 1-4). Each dataset will need its legend removed from the manuscript and added to the corresponding file in a separate tab.

Done.

11) Funding: Please ensure that all funding sources are entered into the manuscript submission system (i.e. please add - 3062/20 grant number for Israeli Science Foundation and the Innovation Seed Grant)

Done.

12) Synopsis:

- Synopsis image: Please include a graphic that summarises the main findings of the manuscript on a glance and upload it as a high-resolution jpeg file 550 pixels wide x (250-400) pixels high.

Done.

- Synopsis text: Please provide a short standfirst (maximum of 300 characters, including space), limit the bullet points to max. 5 and upload it as a separate .doc file. Please write the bullet points to summarise the key NEW findings. They should be designed to be complementary to the abstract - i.e. not repeat the same text. We encourage inclusion of key acronyms and quantitative information (maximum of 30 words / bullet point). Please use the passive voice and upload this as a separate file.

Done.

13) Source Data: Please ensure that a completed Source Data checklist is uploaded (sent to you by my colleague Hannah Sonntag). Please also ensure that the source data are uploaded as a single source data file (zipped) per figure, with the panels clearly visible in the folder structure.

Done.

14) As part of the EMBO Publications transparent editorial process initiative (see our policy here: https://www.embopress.org/transparent-process#Review_Process), Molecular Systems Biology will publish online a Peer Review File (PRF) to accompany accepted manuscripts. This file will be published in conjunction with your paper and will include the anonymous referee reports, your point-by-point response and all pertinent correspondence relating to the manuscript. Let us know whether you agree with the publication of the PRF and as here, if you want to remove or not any figures from it prior to publication. Please note that the Authors checklist will be published at the end of the PRF.

15) Please provide a point-by-point letter INCLUDING my comments as well as the reviewer's reports and your detailed responses (as Word file).

Done.

27th Jun 2024

Manuscript number: MSB-2024-12239RRR

Title: Genome-wide discovery of toxins associated with the extracellular contractile injection system

Dear Dr Levy,

Thank you again for sending us your revised manuscript. We are now satisfied with the modifications made and I am pleased to inform you that your paper has been accepted for publication.

Yours sincerely,

Poonam Bheda, PhD
Scientific Editor
Molecular Systems Biology
